# Swift-SVD: Theoretical Optimality Meets Practical Efficiency in Low-Rank LLM Compression

Ruoling Qi [* 1 2]   Yirui Liu [* ‡ 1]   Xuaner Wu [1]   Xiangyu Wang [1]   Ming Li [3]   Chen Chen [2]
Jian Chen [† 4 5]   Yin Chen [1]   Qizhen Weng [1]

## Abstract

The deployment of Large Language Models is constrained by the memory and bandwidth demands of static weights and dynamic Key-Value cache. SVD-based compression provides a hardware-friendly solution to reduce these costs. However, existing methods suffer from two key limitations: some are suboptimal in reconstruction error, while others are theoretically optimal but practically inefficient. In this paper, we propose Swift-SVD, an activation-aware, closed-form compression framework that simultaneously guarantees theoretical optimum, practical efficiency and numerical stability. Swift-SVD incrementally aggregates covariance of output activations given a batch of inputs and performs a single eigenvalue decomposition after aggregation, enabling training-free, fast, and optimal layer-wise low-rank approximation. We employ effective rank to analyze local layer-wise compressibility and design a dynamic rank allocation strategy that jointly accounts for local reconstruction loss and end-to-end layer importance. Extensive experiments across six LLMs and eight datasets demonstrate that Swift-SVD outperforms state-of-the-art baselines, achieving optimal compression accuracy while delivering 3–70× speedups in end-to-end compression time. Our code is available at https://github.com/hiahei/Swift-SVD.

---

[*]Equal contribution [†]Work done when Jian was at University at Buffalo. [‡]Project lead; <yiruiliu926@gmail.com> [1]Institute of Artificial Intelligence (TeleAI), China Telecom [2]Shanghai Jiao Tong University [3]University of Maryland [4]University at Buffalo [5]Dolby Laboratories. Correspondence to: Jian Chen <Jian.Chen@dolby.com>, Qizhen Weng <wengqzh@chinatelecom.cn>.

*Proceedings of the 43rd International Conference on Machine Learning*, Seoul, South Korea. PMLR 306, 2026. Copyright 2026 by the author(s).

## 1. Introduction

The deployment of large language models (LLMs) is constrained by memory resource requirements during inference. This pressure arises from two sources. First, modern LLMs contain a massive number of parameters that must reside in memory. Second, auto-regressive decoding maintains cached Key–Value (KV) states (Ott et al., 2019), introducing additional runtime memory. Unlike model parameters, these cached representations grow with sequence length, creating a distinct challenge as memory usage and data movement accumulate over time. While hardware advances can help mitigate memory bandwidth limitations (Rhee et al., 2025), its cost and deployment complexity make algorithmic compression approaches an attractive alternative (Shi et al., 2024).

Among algorithmic solutions, post-training compression provides a practical way to reduce resource usage without retraining large models. Existing approaches include quantization (Zhou et al., 2024) and pruning (Guo et al., 2025; Ashkboos et al., 2024), which lower numerical precision and remove parameters, respectively. In contrast, low-rank compression (Ji et al., 2025; Chang et al., 2024) reduces the intrinsic dimensionality of linear layers and can be viewed as a matrix approximation problem that seeks a lower-dimensional projection minimizing reconstruction error under a given objective. This formulation preserves dense operators, maintains compatibility with existing hardware and software stacks, and remains orthogonal to quantization and pruning approaches.

Low-rank compression typically relies on SVD to obtain optimal projections under standard matrix approximation objectives. Early methods directly approximate projection weights for keys and values without explicitly minimizing reconstruction error over data-dependent activations (Chang et al., 2024), which limits their effectiveness under real input distributions. More recent approaches incorporate data dependence (or activation awareness), but often require Cholesky decomposition and/or multiple SVD computations, introducing numerical instability (Wang et al., 2025c; Meyer, 2023; Chen et al., 2021b) and reducing efficiency when scaling to large datasets (Qinsi et al., 2025).

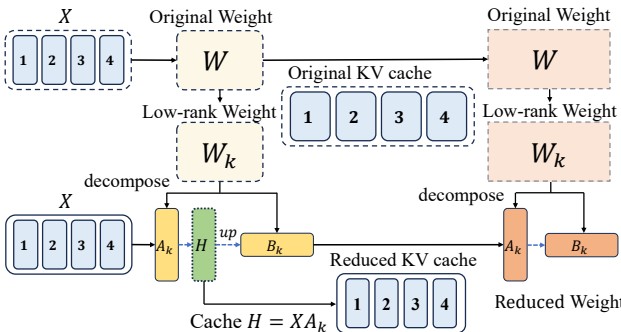

*Figure 1.* Swift-SVD for static weights and KV cache reduction.

Non-uniform compression across layers has also been explored (Wang et al., 2025b; Qinsi et al., 2025); however, the lack of efficient layer-wise loss estimation hinders exhaustive searching for optimal rank allocation. As a result, heuristic strategies are used, which can lead to suboptimal performance that is sometimes worse than uniform rank allocation.

To address these challenges, we propose Swift-SVD, an activation-aware, training-free low-rank compression framework that *jointly reduces the memory footprint of static model weights and the KV cache*, as shown in Figure 1. Swift-SVD provides a direct spectral solution that avoids repeated SVD operations. By performing a single eigenvalue decomposition, it obtains the optimal low-rank projection in closed form, achieving low memory overhead, high efficiency, and strong numerical stability independent of dataset scale or sequence length. The resulting spectral representation also enables fast layer-wise compression loss computation, making grid searches over dynamic rank allocations feasible without relying on heuristics.

To validate the effectiveness of Swift-SVD, we conduct extensive experiments across six LLMs and eight datasets, evaluating end-to-end performance after compression using perplexity and QA accuracy. Swift-SVD consistently outperforms existing low-rank compression baselines, including gradient-based methods (Qinsi et al., 2025). We further demonstrate the time and memory efficiency of Swift-SVD across different compression ratios, as well as its robustness to dataset scale. Our experiments also confirm its numerical stability advantages over methods that rely on repeated SVD. Additionally, we observe that layer compressibility is not solely determined by reconstruction loss. Specifically, we find that the end-to-end layer importance score (Shi et al., 2025) can exhibit a negative correlation with local layer-wise compressibility (see Appendix B.9), which motivates our search strategy for dynamic rank allocation.

Our main contributions are summarized as follows:

- We derive an optimal solution for activation-aware low-

rank compression of LLMs that requires only a single eigenvalue decomposition, improving efficiency, flexibility, and numerical stability.

- We enable fast layer-wise compression using the closed-form formulation, facilitating grid search for optimal dynamic rank allocation beyond uniform compression.

- We reveal a negative correlation between layer importance and compression loss, providing insight into the design of dynamic compression strategies.

- We conduct extensive experiments showing that Swift-SVD outperforms existing low-rank compression baselines on perplexity and QA tasks while maintaining high efficiency across diverse LLMs and datasets.

**Conflict of Interest Disclosure.** The authors declare no financial conflicts of interest relevant to this work.

## 2. Preliminary

### 2.1. Low-Rank Compression for LLMs

As shown in Figure 1, let $X \in \mathbb{R}^{l \times m}$ denote a batch of input activations and $W \in \mathbb{R}^{m \times n}$ the corresponding weight matrix in an LLM. Low-rank compression first computes a rank-k approximation $W_k \in \mathbb{R}^{m \times n}$ with $\text{rank}(W_k) = k$, and then decompose it as $W_k = A_k B_k$, where $A_k \in \mathbb{R}^{m \times k}$ and $B_k \in \mathbb{R}^{k \times n}$. The original weight $W$ is subsequently replaced with $A_k B_k$. This reduces memory usage in two ways: 1) For model weights, the size changes from $m \times n$ to $k(m+n)$, yielding compression whenever $k(m+n) < m \times n$; 2) For KV caches, instead of caching output activations $XW \in \mathbb{R}^{l \times n}$, one can cache intermediate latents $XA_k \in \mathbb{R}^{l \times k}$, which requires less memory whenever $k < n$.

### 2.2. Minimize layer-wise compression loss

For low-rank model compression, a natural question is how to define the compression loss and how to find the optimal $W_k$ that minimizes it. A straightforward choice is to define the loss as the Frobenius norm $||W - W_k||_F$ between the original matrix $W$ and the low-rank approximation matrix $W_k$. Under this definition, the optimal $W_k$ is obtained via the SVD of $W$, where truncating to the top $k$ singular values and their corresponding singular vectors yields the optimal rank-$k$ approximation.

The above loss definition $||W - W_k||_F$ ignores the input activations $X$ entirely. As a result, direct SVD truncation of $W$ leads to significant performance degradation in practice. To address this limitation, recent works adopt an activation-aware (or data-dependent) loss that aims to ensure the output of the compressed LLM closely matches the original output

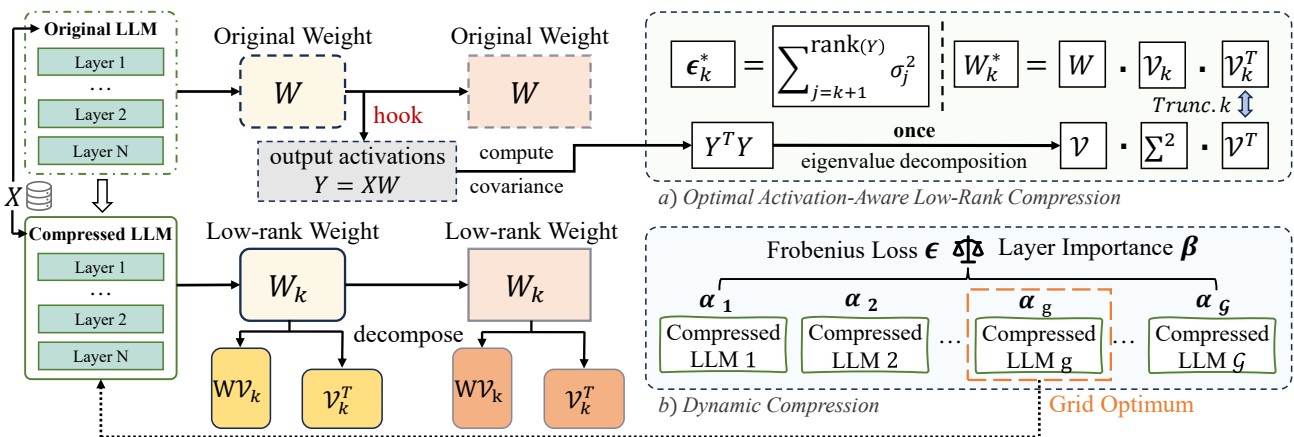

Figure 2. **Overview of Swift-SVD**. *a) Optimal Activation-Aware Low-Rank Compression:* At each transformer layer, Swift-SVD hooks the output activation $Y = XW$ and incrementally aggregates the covariance matrix $Y^T Y$. A single eigenvalue decomposition of this covariance yields the singular values $\Sigma$ and right singular vectors $\mathcal{V}$, from which the optimal activation-aware compression matrix $W_k^*$ and minimal reconstruction loss $\epsilon_k^*$ are derived; *b) Dynamic Compression:* Swift-SVD generates a set of candidate dynamic rank allocation schemes that jointly consider local layer-wise Frobenius loss $\epsilon^*$ and end-to-end layer importance $\beta$. A lightweight grid search is then performed over these candidates—each model is compressed using the optimal solution in *a)* and evaluated on a validation set—to select the configuration that yields the best end-to-end performance.

$Y = XW$. This leads to the following formulation of the compression problem (Chen et al., 2021b; Yuan et al., 2024; Wang et al., 2025c; Qinsi et al., 2025; Wang et al., 2025b):

$$W_k^* = \underset{W_k \in \mathcal{W}}{\arg\min} ||XW - XW_k||_F \qquad (1)$$

$$\epsilon_k^* = ||XW - XW_k^*||_F \qquad (2)$$

where $\mathcal{W}_k = \{W_k \in \mathbb{R}^{m \times n} | \text{rank}(W_k) = k\}$ denote the set of $m \times n$ matrices with rank equal to $k$, and $W_k^*$ and $\epsilon_k^*$ are the optimal compression matrix and the corresponding minimal reconstruction loss for the above optimization problem.

## 3. Method

Swift-SVD is a training-free, activation-aware low-rank compression framework for large language models that combines theoretical optimality with practical efficiency. As illustrated in Figure 2, our method operates in two stages: a) Optimal Activation-Aware Low-Rank Compression; and b) Dynamic compression. Swift-SVD can be seamlessly applied to all types of weight matrices–such as query, key, value, and others–in the same manner. For simplicity, we use the generic notation $W$ throughout our exposition.

### 3.1. Optimal Activation-Aware Low-Rank Compression

This subsection presents the foundation of Swift-SVD: an activation-aware low-rank compression method that computes the optimal weight approximation for any target rank. We first establish a closed-form spectral solution that characterizes the optimal compressed weights and their minimal

reconstruction loss. Then, we describe an efficient incremental algorithm to compute this solution from input activations.

#### 3.1.1. A CLOSED FORM SPECTRAL SOLUTION

Theoretically, Swift-SVD establishes a new theorem that fully characterizes the optimal solution to the activation-aware compression problem defined in (1) and (2). We present the formal statement as follows:

**Theorem 3.1.** *Given input activations $X$ and weight matrix $W$, let $\mathcal{V}$ and $\Sigma$ denote the right singular vectors and singular values of $Y = XW$, respectively. For any $k < rank(Y)$, the optimal solution to the problem defined in (1) and (2) is,*

$$W_k^* = W\mathcal{V}_k\mathcal{V}_k^T, \quad \forall k < rank(Y) \qquad (3)$$

$$\epsilon_k^* = \left(\sum_{j=k+1}^{rank(Y)} \sigma_j^2\right)^{\frac{1}{2}}, \quad \forall k < rank(Y) \qquad (4)$$

*where $\mathcal{V}_k \in \mathbb{R}^{n \times k}$ consists of the top-$k$ right singular vectors corresponding to the $k$ largest singular values.*

*Proof.* Firstly, to show $W_k^*$ is optimal, it suffices to show that: 1) $||XW - XW_k||_F \geq ||XW - XW_k^*||_F, \forall W_k \in \mathcal{W}_k$ and 2) $W_k^* \in \mathcal{W}_k$. For 1) observe that,

$$XW_k^* = XW\mathcal{V}_k\mathcal{V}_k^T \overset{T_1}{=} Y\mathcal{V}_k\mathcal{V}_k^T \overset{T_2}{=} \mathcal{U}\Sigma(\mathcal{V}^T\mathcal{V}_k)\mathcal{V}_k^T$$
$$\overset{T_3}{=} \mathcal{U}\left(\Sigma \begin{bmatrix} I_k \\ 0 \end{bmatrix}\right)\mathcal{V}_k^T = \left(\mathcal{U}\begin{bmatrix} \Sigma_k \\ 0 \end{bmatrix}\right)\mathcal{V}_k^T = \mathcal{U}_k\Sigma_k\mathcal{V}_k^T \qquad (5)$$

Where $I_k$ is a $k$ by $k$ identity matrix. $T_1$ holds because by definition $Y = XW$; $T_2$ holds because $\mathcal{U}\Sigma\mathcal{V}^T$ is the SVD of $Y$; $T_3$ holds because singular vectors form an orthogonal basis, thus $\mathcal{V}_k^T\mathcal{V}_k$ yields a $k$ by $k$ identify matrix. Thus, $XW_k^*$ is precisely the rank-$k$ truncated SVD

of $Y$. By the Eckart-Young-Mirsky theorem (Eckart & Young, 1936), it minimizes $||Y - Y'||_F$ over all $Y'$ with rank$(Y') \leq k$. Moreover, for any $W_k \in \mathcal{W}_k$, we have rank$(XW_k) \leq$ rank$(W_k) = k$. Hence,

$$||XW - XW_k||_F \geq ||XW - XW_k^*||_F, \ \forall \, W_k \in \mathcal{W}_k \quad (6)$$

For 2) note that,

$$\text{rank}(W_k^*) = \text{rank}(W\mathcal{V}_k\mathcal{V}_k^T) \leq \text{rank}(\mathcal{V}_k) \overset{T_4}{=} k \quad (7)$$

$T_4$ holds since $\mathcal{V}_k$ has $k$ orthogonal singular vectors. On the other hand,

$$\text{rank}(W_k^*) \geq \text{rank}(XW_k^*) = \text{rank}(\mathcal{U}_k\Sigma_k\mathcal{V}_k^T) \overset{T_5}{=} k \quad (8)$$

where $T_5$ holds because $\sigma_k > 0$ under the assumption $k <$ rank$(Y)$. Thus, rank$(W_k^*) = k$, which implies $W_k^* \in \mathcal{W}_k$.

Secondly, as $XW_k^*$ is the best $k$-rank approximation of $Y = XW$, it follows direct from the Eckart-Young-Mirsky theorem that the minimal loss is,

$$\epsilon_k^* = ||XW - XW_k^*||_F$$
$$= ||XW - \mathcal{U}_k\Sigma_k\mathcal{V}_k^T||_F = \left(\sum_{j=k+1}^{\text{rank}(Y)} \sigma_j^2\right)^{\frac{1}{2}} \quad (9)$$

$\square$

### 3.1.2. DECOMPOSITION VIA INCREMENTAL AGGREGATION

Theorem 3.1 paves a new pathway to the numerical computation of $W_k^*$ and $\epsilon_k^*$, as it suffices to compute the right singular vectors $\mathcal{V}$ and singular values $\Sigma$ of $XW$. Swift-SVD first computes the covariance matrix of $Y$ as $C = Y^TY$, and then perform an eigenvalue decomposition of $C$ to obtain $\mathcal{V}$ and $\Sigma$. To see why this works, consider the following:

$$Y^TY \overset{T_1}{=} (\mathcal{U}\Sigma\mathcal{V}^T)^T\mathcal{U}\Sigma\mathcal{V}^T = \mathcal{V}\Sigma^T\mathcal{U}^T\mathcal{U}\Sigma\mathcal{V}^T \overset{T_2}{=} \mathcal{V}\Sigma^2\mathcal{V}^T \quad (10)$$

where $T_1$ follows from substituting the SVD of $Y$, and $T_2$ holds because left singular vectors form an orthogonal basis–i.e., $\mathcal{U}^T\mathcal{U} = I$–so $\Sigma^T\mathcal{U}^T\mathcal{U}\Sigma = \Sigma^2$.

This approach enables Swift-SVD to solve the problem efficiently: it requires only a small amount of extra memory to store the $n \times n$ covariance matrix $C$ and a single eigenvalue decomposition, bypassing the need for Cholesky factorization or multiple SVDs. As a result, Swift-SVD is not only fast (see Section 4.2 for experimental results), but also exhibits strong numerical stability (see Section 4.3 for experimental results).

Algorithm 1 presents the pseudo-code of our method. Given $X$ containing $l$ input activation vectors, $\mathbf{y}_t^T\mathbf{y}_t$ is computed and the covariance matrix $C$ is updated. After $l$ iterations, we obtain the full covariance matrix of $Y$. Finally we perform eigen-decomposition on $C$ to obtain the singular values $\Sigma$ and right singular vectors $\mathcal{V}$ of $Y$.

---

**Algorithm 1** Incremental Aggregation Algorithm

**Input:** Input activations $X = [\mathbf{x}_1; ...; \mathbf{x}_l] \in \mathbb{R}^{l \times m}$; LLM model weights $W \in \mathbb{R}^{m \times n}$

**Output:** singular values $\Sigma$ and right singular vectors $\mathcal{V}$ of $Y = XW \in \mathbb{R}^{l \times n}$

1: $C \leftarrow$ zero matrix of size $(n \times n)$
2: **for** $t = 1, \ldots, l$ **do**
3:      $\mathbf{y}_t \leftarrow \mathbf{x}_t W$
4:      $C \leftarrow C + \mathbf{y}_t^T\mathbf{y}_t$
5: **end for**
6: $\mathcal{V}, \Sigma^2, \mathcal{V}^T \leftarrow$ eigen-decomposition$(C)$
7: **return** $\Sigma, \mathcal{V}$

---

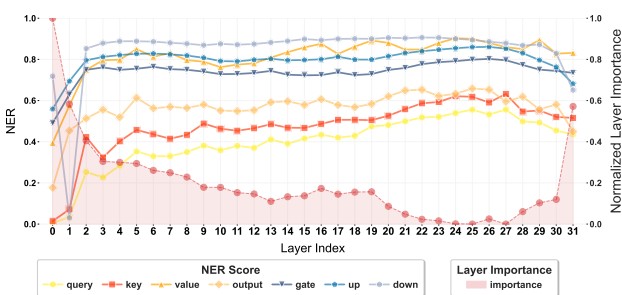

*Figure 3.* Layer-wise NER across distinct modules and layer importance in Mistral-7B with dataset C4.

### 3.2. Dynamic Compression

Uniform rank allocation is often suboptimal because different layers exhibit heterogeneous redundancy. While recent works propose non-uniform strategies—e.g., SVD-LLM v2 (Wang et al., 2025b) uses layer-wise reconstruction loss, and Dobi-SVD (Qinsi et al., 2025) relies on end-to-end training to determine per-layer ranks—they are not optimal and can still under-perform uniform allocation sometimes. To address this, Swift-SVD introduces a novel dynamic rank allocation scheme that jointly considers local layer-wise compressibility and end-to-end compressibility.

### 3.2.1. COMPRESSIBILITY ANALYSIS

Swift-SVD's design of dynamic compression is motivated by the observation that local compressibility (i.e., how well a layer can be approximated at low rank) and end-to-end compressibility (i.e., how much compressing a layer degrades overall model performance) are negatively correlated.

To make this intuition concrete, we employ the effective rank (Roy & Vetterli, 2007) as a quantitative measure of local compressibility. Specifically, given the singular values $\Sigma = \{\sigma_i\}_{i=1}^r$ of a layer's output $Y = XW$–already computed by Algorithm 1–the effective rank is,

$$\text{erank}(\Sigma) = \exp\left(-\sum_{i=1}^r p_i \ln p_i\right). \quad (11)$$

where $p_i = \sigma_i / \sum_{j=1}^r \sigma_j$ is the normalized spectral distribution. A lower effective rank indicates a stronger intrinsic low-rank structure and thus higher local compressibility.

For end-to-end compressibility, we adopt the standard layer importance[1] metric from prior work (Guo et al., 2025; Shi et al., 2025; Song et al., 2025; Men et al., 2024), which estimates the contribution of the i-th layer to the overall LLM performance. Consequently, lower layer importance implies higher end-to-end compressibility.

Figure 3 visualizes the normalized effective rank and normalized layer importance across all seven weight matrices in a representative LLM. Strikingly, the two exhibit a clear negative correlation: layers with high importance tend to have lower effective rank. This validates our core motivation and highlights the need to balance these divergent signals in rank allocation. (Additional results are provided in Appendix B.9.)

### 3.2.2. DYNAMIC COMPRESSION STRATEGY

Following above motivation, Swift-SVD first generates candidate rank allocations based on layer importance and local compression loss, then selects the best one via lightweight validation: each candidate is used to compress the model using the optimal solution in (3), and the resulting models are evaluated on a validation set to identify the one that yields the best end-to-end performance. Thanks to the closed-form spectral solution, this grid search requires no retraining and is easily parallelized.

For clarity, in this subsection we use boldface notation $\boldsymbol{x}_i$ to denote the i-th element of the vector $\boldsymbol{x}$, and let $L$ denote the number of layers in an LLM.

**Generating candidate rank allocations.** The pseudo-code for dynamic rank allocation is shown in **Algorithm** 2. Specifically, given a target compression ratio $\rho \in (0, 1)$, we first compute the uniform rank $\bar{k}$ that would achieve $\rho$ under equal allocation. Then, based on a retention ratio $\delta \in (0, 1]$, we assign a guaranteed minimal rank to each layer as $\boldsymbol{k}_i = \bar{k} \cdot \delta$. Next, we compute a compressibility score for each layer,

$$\boldsymbol{s}_i = (\boldsymbol{\beta}_i)^\alpha \cdot (\log(e + \epsilon^*_{\bar{k},i}))^{1-\alpha} \tag{12}$$

where $\boldsymbol{\beta}_i$ is the layer importance of the i-th layer, min-max normalized to $[0, 1]$ and shifted by 1 (i.e. mapped to $[1,2]$), $\epsilon^*_{\bar{k},i}$ is the minimal reconstruction loss, calculated via (4), for a rank-$\bar{k}$ approximation of i-th layer, $e$ is the base of the natural logarithm, $\alpha \in [0, 1]$ is a hyperparameter balancing the influence of global importance and local compressibility. We then compute the remaining rank budget $b = \bar{k} \times L - \sum_{i=1}^L \boldsymbol{k}_i$, which forms a flexible rank pool. This pool is

---

[1]The referenced code is available at https://github.com/sramshetty/ShortGPT?tab=readme-ov-file.

---

**Algorithm 2** Dynamic Rank Allocation Strategy

**Input:** Frobenius loss $\epsilon$, Layer importance $\boldsymbol{\beta}$, Compression ratio $\rho$, Scaling factor $\alpha$, Retained ratio $\delta$, Size of original weights $m, n$
**Output:** Rank allocation $\boldsymbol{k}$
1: $\boldsymbol{\beta} \leftarrow (\boldsymbol{\beta} - \min(\boldsymbol{\beta}))/(\max(\boldsymbol{\beta}) - \min(\boldsymbol{\beta})) + 1$
2: $\bar{k} = \frac{m \times n}{m+n} \times \rho$         ▷ *Uniform rank*
3: **for all** layer $i \in \{1 \ldots L\}$ **do**
4:    $\boldsymbol{k}_i \leftarrow \bar{k} \cdot \delta$       ▷ *Guaranteed minimal rank*
5:    $\boldsymbol{s}_i \leftarrow (\boldsymbol{\beta}_i)^\alpha \cdot (\log(e + \epsilon_{\bar{k},i}))^{1-\alpha}$
6: **end for**
7: $b \leftarrow \bar{k} \times L - \sum_{i=1}^L \boldsymbol{k}_i$    ▷ *Flexible rank pool*
8: **for all** layer $i \in \{1 \ldots L\}$ **do**
9:    $\boldsymbol{k}_i \leftarrow \boldsymbol{k}_i + [b \cdot (\boldsymbol{s}_i / \sum_{j=1}^L \boldsymbol{s}_j)]$
10: **end for**
11: **return** $\boldsymbol{k}$

---

distributed proportionally to the score $\boldsymbol{s}_i$ as $\boldsymbol{k}_i \leftarrow \boldsymbol{k}_i + [b \cdot (\boldsymbol{s}_i / \sum_{j=1}^L \boldsymbol{s}_j)]$.

**Grid search over candidate rank allocations.** Swift-SVD uses a fixed retention ratio $\delta = 0.5$ and 11 scaling factors $\boldsymbol{\alpha} = [0, 0.1, 0.2, ..., 1]$ to generate 11 candidate rank allocations. For each candidate corresponding to $\boldsymbol{\alpha}_i$, the optimal low-rank approximation of every layer is computed using the closed-form solution in (3). The resulting compressed models are then evaluated on a validation set, and the candidate that yields the best end-to-end performance is selected. (Elaborate experimental analyses of scaling factors $\boldsymbol{\alpha}$ are provided in Appendix A.2.)

## 4. Experiments and Analysis

**Hardware and Software Setup.** All experiments are conducted on a single NVIDIA RTX 5090 GPU (32GB) unless otherwise specified, using PyTorch 2.8.0 and Hugging Face Transformers 4.57.3. All evaluations are run in inference mode without gradient computation.

**Baselines.** We evaluate the performance of Swift-SVD against five state-of-the-art SVD-based LLM compression methods: FWSVD (Hsu et al., 2022), ASVD (Yuan et al., 2024), SVD-LLM (Wang et al., 2025c), SVD-LLM v2 (Wang et al., 2025b), and Dobi-SVD (Qinsi et al., 2025).

**Models and Datasets.** To evaluate robustness across architectures and scales, we experiment on a diverse suite of LLMs: LLaMA-7B, LLaMA2-7B (Touvron et al., 2023), OPT-6.7B (Zhang et al., 2022), Mistral-7B (Jiang et al., 2023), and Qwen3 (4B, 8B) (Team, 2025). We assess performance using nine standard benchmarks: WikiText-2 (Merity et al., 2016), C4 (Raffel et al., 2023), and Alpaca (Taori et al., 2023) for language modeling (perplexity), and OpenBookQA (Mihaylov et al., 2018), Wino-

*Table 1.* Performance comparison of LLaMA-7B on language modeling and zero-shot tasks. Baseline results are reported from their original papers. We mark the **best** and second-best results. Regarding specific variants: **Swift-SVD** employs a uniform rank allocation strategy, while **Swift-SVD\*** utilizes our proposed dynamic compression. **SVD-LLM(W)** denotes the uniform compression of SVD-LLM with truncation-aware whitening only, and **Dobi-SVD(w/o) / (w)** represent the method without / with dynamic rank allocation.

| Ratio(MEM.) | Method | PPL (↓) | | Accuracy (↑) | | | | | | |
| --- | --- | --- | --- | --- | --- | --- | --- | --- | --- | --- |
| | | WikiText-2 | C4 | ARC_e | PIQA | Openb. | WinoG. | HellaS. | MathQA | Avg. |
| 1.0(12.6GB) | Baseline | 5.68 | 7.34 | 0.76 | 0.79 | 0.34 | 0.70 | 0.57 | 0.27 | 0.57 |
| **0.8** (10.1 GB) | FWSVD | 1727 | 1511 | 0.11 | 0.10 | 0.09 | 0.05 | 0.08 | 0.05 | 0.08 |
| | ASVD | 11.14 | 15.93 | 0.53 | 0.68 | 0.29 | 0.64 | 0.41 | 0.17 | 0.45 |
| | SVD-LLM(W) | 7.94 | 15.84 | 0.62 | 0.71 | **0.31** | 0.61 | 0.45 | 0.21 | 0.49 |
| | Dobi-SVD(w/o) | 8.87 | 10.91 | - | - | - | - | - | - | - |
| | Dobi-SVD(w) | 8.54 | **10.01** | 0.63 | 0.72 | 0.30 | 0.62 | 0.46 | 0.20 | 0.49 |
| | **Swift-SVD** | 7.91 | 11.42 | 0.64 | 0.73 | 0.26 | 0.68 | 0.47 | 0.23 | 0.50 |
| | **Swift-SVD\*** | **7.84** | 11.15 | **0.65** | **0.73** | 0.27 | **0.68** | **0.48** | **0.23** | **0.51** |
| **0.6** (7.7 GB) | FWSVD | 18156 | 12847 | 0.05 | 0.05 | 0.06 | 0.02 | 0.00 | 0.03 | 0.04 |
| | ASVD | 1407 | 1109 | 0.11 | 0.13 | 0.08 | 0.09 | 0.08 | 0.08 | 0.10 |
| | SVD-LLM(W) | 13.73 | 75.42 | 0.33 | 0.63 | **0.25** | 0.55 | **0.40** | 0.12 | 0.38 |
| | Dobi-SVD(w/o) | 14.96 | 24.60 | - | - | - | - | - | - | - |
| | Dobi-SVD(w) | 13.54 | 23.54 | 0.45 | 0.64 | 0.22 | 0.58 | 0.36 | 0.18 | 0.41 |
| | **Swift-SVD** | 13.42 | 23.32 | 0.49 | 0.66 | 0.21 | 0.62 | 0.37 | 0.22 | 0.43 |
| | **Swift-SVD\*** | **13.29** | **21.92** | **0.51** | **0.67** | 0.23 | **0.62** | 0.38 | **0.22** | **0.44** |
| **0.4** (5.3 GB) | FWSVD | 32194 | 29292 | 0.02 | 0.02 | 0.06 | 0.01 | 0.01 | 0.03 | 0.03 |
| | ASVD | 57057 | 43036 | 0.04 | 0.08 | 0.05 | 0.06 | 0.09 | 0.05 | 0.06 |
| | SVD-LLM(W) | 66.62 | 471.83 | 0.05 | 0.21 | 0.10 | 0.17 | 0.10 | 0.04 | 0.11 |
| | Dobi-SVD(w/o) | 58.02 | 145.41 | - | - | - | - | - | - | - |
| | Dobi-SVD(w) | **46.18** | 190.62 | 0.25 | 0.51 | 0.14 | 0.48 | 0.24 | 0.15 | 0.30 |
| | Swift-SVD | 64.16 | 143.74 | 0.29 | 0.56 | 0.16 | 0.52 | 0.27 | 0.21 | 0.34 |
| | Swift-SVD\* | 62.32 | **137.77** | **0.30** | **0.57** | **0.16** | **0.53** | **0.28** | **0.21** | **0.34** |

*Table 2.* Cross-model compression performance under 0.8 compression ratio. Baseline results are reported from their original papers. Results show PPL on WikiText-2/C4 and average accuracy on six common sense reasoning benchmarks.

| Model | OPT-6.7B | | | LLAMA 2-7B | | | MISTRAL-7B | | |
| --- | --- | --- | --- | --- | --- | --- | --- | --- | --- |
| | Perplexity↓ | | Acc↑ | Perplexity↓ | | Acc↑ | Perplexity↓ | | Acc↑ |
| **Method** | Wiki2 | C4 | Avg. | Wiki2 | C4 | Avg. | Wiki2 | C4 | Avg. |
| Original | 10.86 | 12.52 | 0.52 | 5.47 | 9.30 | 0.57 | 5.25 | 9.28 | 0.61 |
| ASVD | 82.04 | 102 | 0.32 | 10.10 | 24.02 | 0.36 | 13.72 | 23.34 | 0.32 |
| SVD-LLM (W) | 16.04 | 21.27 | 0.41 | 8.50 | 12.69 | 0.53 | 10.21 | 13.17 | 0.42 |
| Swift-SVD | 12.12 | 17.93 | 0.50 | 8.41 | 12.54 | 0.56 | 7.40 | 12.80 | 0.54 |
| **Swift-SVD\*** | **11.65** | **13.68** | **0.51** | **8.27** | **12.35** | **0.56** | **6.63** | **11.08** | **0.55** |

Grande (Sakaguchi et al., 2019), HellaSwag (Zellers et al., 2019), ARC-Easy (Clark et al., 2018), PIQA (Bisk et al., 2020), and MathQA (Amini et al., 2019) for zero-shot common sense reasoning.

**Evaluation metrics.** We evaluate model performance using perplexity (PPL) (Bengio et al., 2003) and zero-shot accuracy, computational efficiency via compression time (in seconds), inference efficiency through memory usage (in GB) and throughput (in tokens/second), and numerical stability by reconstruction loss, computed from (2).

### 4.1. Performance Analysis

**Compression with Different Methods.** First, we benchmark Swift-SVD against state-of-the-art SVD-based compression methods across varying compression ratios. To ensure a fair comparison, baseline results are directly cited from their original papers, and Swift-SVD is evaluated under the same experimental protocol as each corresponding baseline, including calibration dataset and sample size. We evaluate performance using PPL and zero-shot accuracy on a diverse set of benchmarks. Evidently in Table 1, Swift-SVD achieves a comprehensive improvement over baselines, securing the highest average accuracy across all compression levels while yielding the lowest PPL in the majority of cases. This robustness is particularly pronounced under aggressive compression, where our method maintains high performance. Our dynamic allocation strategy ensures consistent gains and avoids the instability observed in Dobi-SVD. For example, on the C4 dataset at a compression ratio of 0.4, Dobi-SVD(w) achieves a PPL of 190.62, which is significantly worse than the uniform counterpart Dobi-SVD(w/o) that achieves 145.41.

The low-rank matrices produced by Swift-SVD are directly compatible with LoRA fine-tuning as initialization, following the same paradigm as SVD-LLM. The quality of initialization matters for recovery: as reported in SVD-LLM's original paper, at compression ratio 0.4 on LLaMA-7B, SVD-LLM achieves only 0.11 average accuracy after compression, and even after LoRA fine-tuning recovers to only 0.30—still below Swift-SVD's 0.34 achieved by training-free compression alone (Table 1). This suggests that a poor initialization can fundamentally limit LoRA fine-tuning effectiveness, and a stronger initialization such as Swift-SVD is expected to yield further gains when combined with LoRA. We leave this as future work. We additionally pro-

*Table 3.* Cross-domain PPL of LLaMA-7B. *Original* uses the evaluation set for calibration; *PPL* uses C4-only.

| | nsamples = 256 | | | | nsamples = 2048 | | |
|---|---|---|---|---|---|---|---|
| Ratio | Datasets | Original | PPL | Ratio | Datasets | Original | PPL |
| 0.8 | C4 | 11.42 | 11.42 | 0.8 | C4 | 11.34 | 11.34 |
| | WikiText 2 | 7.86 | 11.33 | | WikiText 2 | 7.81 | 11.31 |
| | Alpaca | 8.49 | 10.51 | | Alpaca | 7.84 | 10.38 |
| 0.6 | C4 | 23.17 | 23.17 | 0.6 | C4 | 22.37 | 22.37 |
| | WikiText 2 | 13.42 | 37.00 | | WikiText 2 | 13.37 | 35.02 |
| | Alpaca | 12.31 | 16.40 | | Alpaca | 9.90 | 16.82 |
| 0.4 | C4 | 137.01 | 137.01 | 0.4 | C4 | 136.21 | 136.21 |
| | WikiText 2 | 64.16 | 285.87 | | WikiText 2 | 63.01 | 267.98 |
| | Alpaca | 33.97 | 78.27 | | Alpaca | 24.18 | 77.48 |

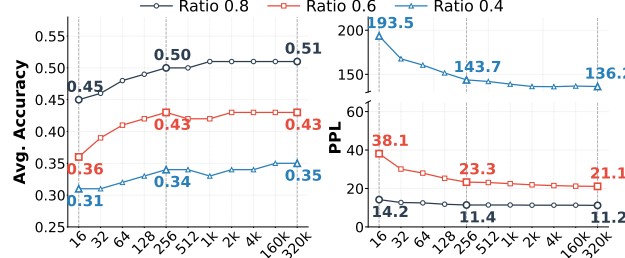

*Figure 4.* Impact of calibration sample size $N$ on model performance. We report average accuracy on zero-shot tasks (left) and PPL on C4 (right) across three compression ratios.

vide comparisons on Qwen3-4B under both uniform and dynamic compression in Appendix B.3, evaluate scalability on Qwen-32B in Appendix B.4, and compare with Basis Sharing (Wang et al., 2025a) in Appendix B.2. Qualitative examples of text generated by compressed models are provided in Appendix B.10.

**Performance across Different LLMs.** To evaluate the generalization capability of Swift-SVD across diverse architectures under uniform compression, we benchmark its performance against SVD-LLM(W) and ASVD on OPT-6.7B, LLaMA2-7B, and Mistral-7B. As presented in Table 2, both Swift-SVD and Swift-SVD* outperform the baselines across all three models, demonstrating superior stability and universality across different LLM architectures.

**Cross-dataset Generalization.** To validate the effectiveness of the **activation-aware mechanism** in Swift-SVD, we conducted a cross-domain evaluation. We utilized the C4 dataset for calibration and evaluated the compressed model across three distinct domains.

As illustrated in Table 3, significant PPL degradation is observed when the calibrated model with C4 is applied to WikiText-2 or Alpaca, demonstrating that Swift-SVD is acutely activation-aware. Further task-specific accuracy analysis under different calibration strategies is provided in Appendix B.1.

**Impact of Aware Samples Size.** We investigate the sensitivity of Swift-SVD to the calibration sample size $N$ in Figure 4. The results demonstrate a clear trend: performance improves rapidly in the low-sample regime but shows diminishing returns as $N$ increases further. While larger samples continue to yield marginal gains, we adopt the standard setting of $N = 256$ to ensure a fair comparison with baselines. Details on calibration data construction are provided in Appendix A.1.

### 4.2. Computational Efficiency

**End-to-end Compression Time.** We compare the end-to-end compression time of Swift-SVD against baseline methods under a uniform compression strategy across three

compression ratios: $\rho = 0.8$, $\rho = 0.6$ and $\rho = 0.4$. The results, shown in Table 4, demonstrate that Swift-SVD achieves substantial speedups over all baselines–up to $3.8\times$ compared to SVD-LLM(W), and up to $76.9\times$ compared to Dobi-SVD(w/o). This efficiency stems from two key advantages: First, Algorithm 1 is highly efficient—it only requires incremental aggregation of the activation covariance matrix followed by a single eigen-decomposition. In contrast, baseline methods (notably Dobi-SVD) perform a full SVD for each input sample, which is computationally prohibitive; Second, Swift-SVD computes the entire optimal solution spectrum, as shown in (3) in one pass. Consequently, for any subsequent compression ratio (e.g., $\rho = 0.6$ or $\rho = 0.4$), no re-compression is needed. As a result, the end-to-end compression time collapses to the time required to load the compressed weights into memory.

*Table 4.* End-to-end compression latency (in seconds) evaluated on the C4 dataset. We report the total time required to compress the complete model.

| Samples | Method | $\rho = 0.8$ | $\rho = 0.6$ | $\rho = 0.4$ | Total (s) | Speedup |
|---|---|---|---|---|---|---|
| 16 | Dobi-SVD(w/o) | 960 | 650 | 373 | 1,983 | 1.0× |
| | SVD-LLM (W) | 542 | 489 | 503 | 1,534 | 1.3× |
| | **Swift-SVD** | **342** | **146** | **133** | **621** | **3.2×** |
| 64 | Dobi-SVD(w/o) | 3,823 | 2,551 | 1,508 | 7,882 | 1.0× |
| | SVD-LLM (W) | 555 | 530 | 565 | 1,650 | 4.8× |
| | **Swift-SVD** | **346** | **158** | **150** | **654** | **12.1×** |
| 256 | Dobi-SVD(w/o) | 15,269 | 10,468 | 5,966 | 31,703 | 1.0× |
| | SVD-LLM (W) | 757 | 734 | 722 | 2,213 | 14.3× |
| | **Swift-SVD** | **453** | **152** | **148** | **753** | **42.1×** |
| 512 | Dobi-SVD(w/o) | 30,862 | 20,984 | 11,795 | 63,641 | 1.0× |
| | SVD-LLM (W) | 1,080 | 1,069 | 1,063 | 3,212 | 19.8× |
| | **Swift-SVD** | **540** | **157** | **130** | **827** | **76.9×** |

**Inference Speedup and Memory Reduction.** To quantify the acceleration and memory reduction achieved by Swift-SVD, we evaluate the inference throughput (tokens per second) and peak memory footprint of LLaMA-7B on a single NVIDIA 5090 GPU. We report performance under various batch sizes and sequence lengths (see Appendix B.6), and compare serving throughput across methods in Appendix B.7. Throughput results at longer generation lengths are provided in Appendix B.8. Furthermore, we analyze the scaling behavior of the total model size, comprising both compressed weights and reduced KV cache overhead,

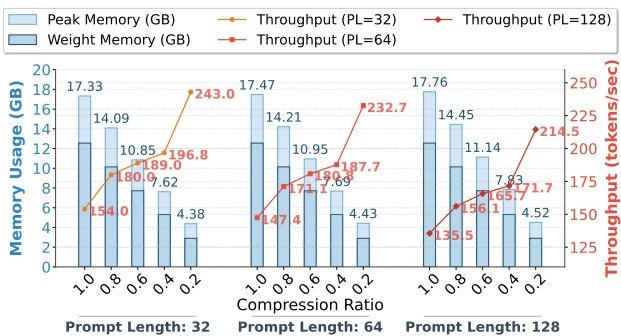

*Figure 5.* Throughput improvement and memory efficiency under batch size of 16. The generated sequence length is 1024.

across various compression ratios. Our experimental results shown in Figure 5 demonstrate that as the compression ratio decreases, our method significantly enhances inference efficiency by boosting throughput and alleviating HBM pressure.

### 4.3. Numerical Stability

To evaluate the numerical stability of Swift-SVD and baseline methods, we generate random matrices of varying sizes to simulate input activations $X$ and weights $W$ across a range of dimensions. With a fixed compression ratio of 0.6, we compare the reconstruction loss of each method against the theoretical minimum loss. Although both SVD-LLM and Dobi-SVD are designed to be theoretically optimal, they all suffer from numerical instabilities, resulting in reconstruction losses consistently higher than the theoretical minimum. In contrast, Swift-SVD achieves near-perfect alignment with the theoretical optimum across all scales. These results confirm that Swift-SVD provides a more numerically robust solution.

*Table 5.* Reconstruction loss and absolute error for randomly generated matrices of varying shapes under ratio of 0.6 (FP32).

| **Input $\times$ Weight** | $[128 \times 128]^2$ | $[1024 \times 1024]^2$ |
|---|---|---|
| Minimum | 126.2506 | 841.1812 |
| SVD-LLM | 126.7102 (+0.4596) | 854.4129 (+13.2317) |
| Dobi-SVD | 128.5441 (+2.2935) | 874.7711 (+33.5899) |
| Swift-SVD | 126.2506 (+0.0000) | 841.1812 (+0.0000) |

| **Input $\times$ Weight** | $[2048 \times 2048]^2$ | $[4096 \times 4096]^2$ |
|---|---|---|
| Minimum | 1657.4801 | 3308.6428 |
| SVD-LLM | 1686.1804 (+28.7003) | 3365.6499 (+57.0071) |
| Dobi-SVD | 1724.2129 (+66.7328) | 3442.5242 (+133.8814) |
| Swift-SVD | 1657.4801 (+0.0000) | 3308.6428 (+0.0000) |

### 4.4. Ablation Study

To evaluate the contribution of each component, we compare our proposed Swift-SVD* against several variants: (1) Swift-SVD, a baseline that assigns a uniform rank; (2) Swift-SVD(C), which mirrors the dynamic rank allocation strategy

*Table 6.* PPL ($\downarrow$) of compressed LLMs of different allocation strategies across various models under the ratio of 0.8 on C4.

| Strategy | LLaMA-7B | LLaMA-2-7B | OPT-6.7B | Mistral-7B |
|---|---|---|---|---|
| Swift-SVD | 11.42 | 12.54 | 17.93 | 12.80 |
| Swift-SVD(C) | 16.04 | 22.16 | 20.69 | 22.87 |
| Swift-SVD(I) | 14.88 | 17.30 | 18.94 | 22.50 |
| Swift-SVD$^{\dagger}$(C) | 11.78 | 13.74 | 15.66 | 11.67 |
| Swift-SVD$^{\dagger}$(I) | 11.73 | 13.27 | 14.74 | 11.48 |
| **Swift-SVD*** | **11.15** | **12.35** | **13.68** | **11.08** |

of SVD-LLM v2 (Wang et al., 2025b) by allocating ranks solely based on Frobenius loss, and Swift-SVD(I), which allocate ranks based on layer importance without any preserved ratio; and (3) Swift-SVD$^{\dagger}$(C) and Swift-SVD$^{\dagger}$(I), which incorporate a fixed preserved ratio of $\delta = 0.5$. As shown in Table 6, unrestricted dynamic allocation proves detrimental. This confirms that relying exclusively on Frobenius loss or layer importance risks undermining the module's essential representation capacity by over-compressing specific modules. In contrast, enforcing a fixed preserved ratio (e.g., $\delta = 0.5$) acts as a stabilizer, effectively mitigating error propagation and reversing this degradation. Ultimately, Swift-SVD* achieves the best performance by synergizing this basic structural preservation with fine-grained redundancy exploitation. We further analyze the sensitivity of hyperparameters $\alpha$ and $\delta$ in Appendix A.2, and visualize the singular value distribution motivating our dynamic allocation design in Appendix B.5.

## 5. Related Work

**Rank Analysis in Language Models.** Early work has investigated the relationship between the rank of transformer weights or representations and model performance, seeking either to leverage low-rank structure for efficiency (Chen et al., 2021a; Hsu et al., 2022; Hajimolahoseini et al., 2022; Li et al., 2023), to prevent rank collapse that limits expressivity (Dong et al., 2021; Noci et al., 2022; Yaras et al., 2024), or to maximize rank utilization for enhanced modeling capacity (Bhojanapalli et al., 2020; Boix-Adsera et al., 2023). With the growing use of LLMs, research has turned to their inherent low-rank properties. LoRA (Hu et al., 2022) leverages this structure during fine-tuning, showing that many weight updates lie in low-dimensional subspaces. Loki (Singhania et al., 2024) examined the key representations in attention layers and found that they often reside in lower-dimensional subspaces across models and datasets, which can be used for efficient sparse attention. These directions also motivated growing efforts on compression (Shi et al., 2024) to address the deployment bottleneck in reading and storing the model and KV cache (Yu et al., 2022).

**Low-Rank Model Compression.** Recent studies explore low-rank joint KV cache compression to facilitate scalable

inference. MHA2MLA (Ji et al., 2025) and PALU (Chang et al., 2024) employ SVD to reformulate Multi-Head Attention (MHA) into Multi-head Latent Attention (MLA). While effective, these weight-only approaches overlook the intrinsic low-rank properties of activations; as evidenced by (Yu & Wu, 2023), transformer weights typically exhibit higher rank than their corresponding activations, suggesting that activation-aware compression is more effective. In this direction, DRONE (Chen et al., 2021b) establishes a closed-form solution for intermediate representations, yet its heavy reliance on caching full activations poses significant memory constraints for LLMs. FWSVD (Hsu et al., 2022) incorporates Fisher information for importance weighting, albeit at the cost of expensive gradient computations. ASVD (Yuan et al., 2024) attempts to normalize activation impact via diagonal scaling but fails to reach the theoretical minimum truncation loss. More recently, KV-CoRE (Chen et al., 2026; Chen et al.), Dobi-SVD (Qinsi et al., 2025), and SVD-LLM (Wang et al., 2025b) have achieved theoretical optimum. However, KV-CoRE focuses on compressibility analysis without practical compressibility scheme. Dobi-SVD relies on Incremental PCA and gradient-based training, a combination that leads to numerical instability and renders the implementation both time-consuming and memory-intensive. SVD-LLM utilizes Cholesky decomposition, which necessitates that the matrices remain positive-definite, a condition that is challenging in diverse activation distributions. Layer importance scores have also been leveraged to guide compression decisions. ShortGPT (Men et al., 2024) measures each layer's contribution via Block Influence (BI) scores and removes entire redundant layers. MoDeGPT (Che et al., 2024) applies BI scores for per-block rank allocation in a modular decomposition framework. In contrast, Swift-SVD retains all layers and jointly considers both layer-wise reconstruction loss and layer importance for fine-grained per-matrix rank allocation; as shown in our ablation (Table 6), layer importance alone proves insufficient for low-rank compression. These limitations necessitate a unified framework that attains the theoretical optimum of truncation error without compromising computational efficiency or numerical stability.

## 6. Conclusion

We propose Swift-SVD, a training-free activation-aware compression framework that reconciles theoretical optimum with practical efficiency. By formulating compression as a closed-form eigenvalue decomposition problem, Swift-SVD eliminates the numerical instability and computational bottlenecks. Swift-SVD further exploits layer-wise compressibility and importance for dynamic rank allocation strategy. Extensive experiments demonstrate that Swift-SVD delivers $3$–$70\times$ end-to-end compression speedups while maintaining state-of-the-art performance across diverse architectures.

Furthermore, as Swift-SVD's compressed model is directly compatible with LoRA fine-tuning as initialization, combining the two is a promising direction for further performance recovery, which we leave as future work.

## Acknowledgements

We thank the anonymous reviewers for their thoughtful comments and constructive suggestions, which helped improve this work. We also thank Zhuoran Wang, Jiayu Qin, Meng Wang, and Daxiang Kang for their valuable discussions, constructive feedback, and technical support throughout the development of this work. Their insights contributed to shaping the ideas and experiments presented in this paper.

## Impact Statement

This paper presents work whose goal is to advance the field of Machine Learning. There are many potential societal consequences of our work, none which we feel must be specifically highlighted here.

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

# A. Additional Experimental Details and Analysis

In this section, we provide additional details on calibration data and an analysis of the hyperparameters used in our dynamic compression experiments.

## A.1. Calibration Data

During the incremental statistic aggregation phase, we randomly select $N = 256$ calibration samples. For text datasets such as WikiText-2 and C4, each sample is processed with a fixed sequence length of 2048 tokens. In contrast, for conversational datasets like Alpaca and zero-shot common sense reasoning benchmarks (e.g., PIQA), samples are constructed by formatting individual raw data entries according to their official prompt templates and concatenating them as discrete, individually separated instances. This ensures that each data entry remains distinct, although it precludes a guaranteed uniform sequence length across all calibration instances. Under such variable-length conditions, the SVD-LLM method is highly susceptible to producing non-positive-definite matrices $X^T X$, frequently leading to numerical instability or complete decomposition failure. Swift-SVD effectively circumvents these limitations by utilizing a direct closed-form eigenvalue decomposition, ensuring robust performance regardless of sequence irregularity or the presence of padding/formatting boundaries.

## A.2. Hyperparameter Analysis in Dynamic Compression

In our dynamic rank allocation strategy, there are two primary hyperparameters: the **scaling factor** $\alpha \in [0, 1]$ and the **preserved ratio** $\delta \in [0, 1]$. 1) $\alpha$ serves as a critical hyperparameter that modulates the trade-off between reconstruction loss $\epsilon$ and layer importance $\beta$. Specifically, $\alpha = 0$ reduces the strategy to a purely loss-driven allocation, whereas $\alpha = 1$ prioritizes layer importance exclusively. 2) $\delta$ acts as a performance stabilizer. Without a preserved ratio $\delta = 0$, unrestricted dynamic allocation can inadvertently destroy the representational capacity of critical layers, leading to significant degradation. Introducing a fixed baseline (e.g., $\delta = 0.5$) effectively mitigates error propagation and reverses this trend. However, excessively large $\delta$ values constrain the flexible budget available for reallocation, slightly reducing the optimization gains from dynamic redundancy exploitation. Notably, when $\delta = 1.0$, the rank allocation $\boldsymbol{k}$ matches the uniform target rank $\bar{k}$ for all modules, rendering the strategy equivalent to uniform compression.

Swift-SVD eliminates the overhead of redundant SVD operations by performing a single eigenvalue decomposition of activation statistics and caching the resulting spectral components ($\Sigma$ and $\mathcal{V}$) for subsequent reuse. This allows for direct truncation and model reconstruction based on dynamic rank assignments, facilitating the rapid acquisition of the full compressed model. Such computational efficiency provides the necessary foundation for expanding our search grid over the hyperparameter space. By defining a comprehensive and reasonable search grid, we can effectively identify the optimal dynamic allocation configuration that maximizes performance of the compressed model within the compressed candidates.

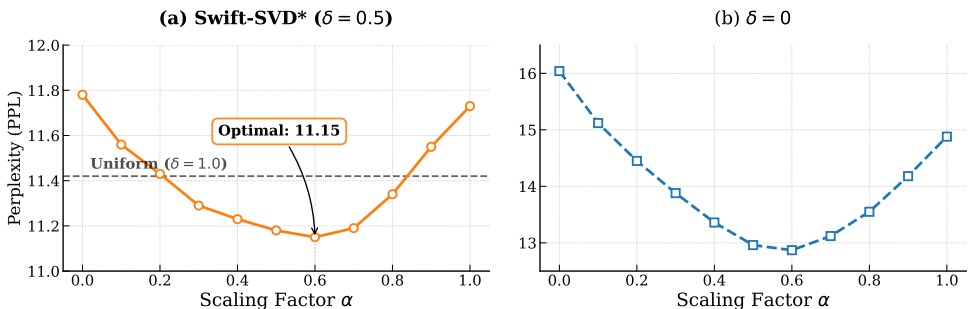

*Figure A.1.* Impact of hyperparameters $\alpha$ and $\delta$ on model performance for dynamic compression. Both configurations exhibit a U-shaped PPL curve.

# B. Additional Experiment Results

## B.1. Accuracy on Specific Task

To evaluate the impact of calibration data on downstream task performance, we compare three calibration strategies across seven zero-shot common sense reasoning. As shown in Table B.1, we define three settings: *C4* (calibrated on general C4 data), *Each* (calibrated specifically on the validation set of each respective task), and *All* (calibrated on a unified mixture of

all validation datasets). We observe a clear hierarchy: *Each* $\gtrsim$ *All* > *C4*.

**Domain Specificity (*Each*):** Achieves the highest accuracy, confirming that task-aligned calibration is optimal for preserving task-critical features.

**Robust Aggregation (*All*):** Matches the oracle *Each* performance closely, proving that Swift-SVD effectively fuses distinct feature subspaces into a single projection matrix to some extent.

**Generalization Limit (*C4*):** The notable accuracy drop illustrates the inadequacy of general-purpose data in capturing fine-grained patterns required for specific tasks.

In summary, while domain-specific calibration is optimal, Swift-SVD demonstrates strong capacity for unified calibration, allowing a single compressed model to serve multiple downstream tasks effectively when provided with mixed calibrations.

*Table B.1.* Benchmark results categorized by compression ratios and aware methods. The Original row is highlighted with gray text.

| Model | Ratio | Aware | ARC_e | ARC_c | PIQA | Openb. | WinoG. | mathQA | HellaS. | Avg |
|---|---|---|---|---|---|---|---|---|---|---|
| | Original | **Object** | 0.76 | 0.42 | 0.79 | 0.34 | 0.70 | 0.27 | 0.67 | 0.56 |
| | | C4 | 0.64 | 0.33 | 0.73 | 0.26 | 0.67 | 0.23 | 0.47 | 0.48 |
| | 0.8 | All | 0.70 | 0.38 | 0.76 | 0.29 | 0.68 | 0.26 | 0.49 | 0.51 |
| | | Each | 0.71 | 0.38 | 0.76 | 0.30 | 0.68 | 0.26 | 0.49 | 0.51 |
| Llama-7B | | C4 | 0.49 | 0.25 | 0.66 | 0.21 | 0.62 | 0.22 | 0.37 | 0.40 |
| | 0.6 | All | 0.61 | 0.31 | 0.72 | 0.25 | 0.62 | 0.25 | 0.40 | 0.45 |
| | | Each | 0.63 | 0.32 | 0.72 | 0.26 | 0.62 | 0.26 | 0.41 | 0.46 |
| | | C4 | 0.29 | 0.20 | 0.56 | 0.16 | 0.52 | 0.21 | 0.27 | 0.32 |
| | 0.4 | All | 0.46 | 0.23 | 0.63 | 0.18 | 0.53 | 0.22 | 0.30 | 0.36 |
| | | Each | 0.53 | 0.25 | 0.64 | 0.20 | 0.56 | 0.25 | 0.31 | 0.39 |
| | Original | | 0.81 | 0.50 | 0.75 | 0.30 | 0.66 | 0.46 | 0.52 | 0.57 |
| | | C4 | 0.69 | 0.37 | 0.70 | 0.26 | 0.61 | 0.29 | 0.42 | 0.48 |
| | 0.8 | All | 0.71 | 0.41 | 0.71 | 0.27 | 0.61 | 0.39 | 0.43 | 0.50 |
| | | Each | 0.73 | 0.42 | 0.74 | 0.27 | 0.62 | 0.44 | 0.44 | 0.52 |
| Qwen3-4B | | C4 | 0.54 | 0.26 | 0.65 | 0.23 | 0.59 | 0.23 | 0.35 | 0.41 |
| | 0.6 | All | 0.58 | 0.32 | 0.69 | 0.24 | 0.60 | 0.32 | 0.36 | 0.45 |
| | | Each | 0.62 | 0.33 | 0.71 | 0.25 | 0.60 | 0.36 | 0.38 | 0.46 |
| | | C4 | 0.28 | 0.19 | 0.56 | 0.14 | 0.49 | 0.20 | 0.27 | 0.30 |
| | 0.4 | All | 0.44 | 0.23 | 0.62 | 0.17 | 0.52 | 0.27 | 0.28 | 0.36 |
| | | Each | 0.49 | 0.23 | 0.64 | 0.19 | 0.54 | 0.31 | 0.30 | 0.39 |
| | Original | | 0.83 | 0.55 | 0.76 | 0.31 | 0.68 | 0.50 | 0.57 | 0.60 |
| | | C4 | 0.75 | 0.44 | 0.73 | 0.27 | 0.64 | 0.32 | 0.47 | 0.52 |
| | 0.8 | All | 0.76 | 0.45 | 0.74 | 0.29 | 0.65 | 0.45 | 0.47 | 0.54 |
| | | Each | 0.77 | 0.45 | 0.75 | 0.30 | 0.65 | 0.48 | 0.48 | 0.55 |
| Qwen3-8B | | C4 | 0.57 | 0.32 | 0.68 | 0.23 | 0.60 | 0.23 | 0.38 | 0.43 |
| | 0.6 | All | 0.68 | 0.37 | 0.71 | 0.26 | 0.61 | 0.39 | 0.39 | 0.49 |
| | | Each | 0.70 | 0.38 | 0.71 | 0.26 | 0.62 | 0.43 | 0.41 | 0.50 |
| | | C4 | 0.36 | 0.19 | 0.59 | 0.14 | 0.54 | 0.22 | 0.30 | 0.34 |
| | 0.4 | All | 0.53 | 0.26 | 0.65 | 0.21 | 0.54 | 0.29 | 0.32 | 0.40 |
| | | Each | 0.56 | 0.28 | 0.66 | 0.22 | 0.55 | 0.36 | 0.33 | 0.42 |

## B.2. Comparison with Basis Sharing

We compare Swift-SVD against Basis Sharing (Wang et al., 2025a), a concurrent method that addresses a similar low-rank compression problem. Basis Sharing results are reported directly from their original paper under the FP32, LLaMA-7B setting, and Swift-SVD is evaluated under the same experimental protocol, including calibration dataset (C4) and sample size. Table B.2 reports perplexity and zero-shot accuracy results. Swift-SVD achieves lower perplexity on C4 and higher average accuracy across most compression ratios. On WikiText-2 at ratio 0.6, Basis Sharing achieves comparable perplexity, while Swift-SVD recovers better on downstream task accuracy.

## B.3. Comparison on Qwen3-4B

We compare Swift-SVD with SVD-LLM(W) and Dobi-SVD(w/o) on Qwen3-4B, a recent architecture not covered by existing baselines in the main text. All results are training-free and obtained without any fine-tuning or post-compression training. MoDeGPT (Che et al., 2024) is excluded as it does not natively support the Qwen3 architecture. Table B.3 reports zero-shot accuracy under both uniform and dynamic rank allocation. Baseline results are reported from their original papers.

*Table B.2.* Perplexity and zero-shot accuracy comparison between Basis Sharing and Swift-SVD on LLaMA-7B (FP32). The Original row is highlighted with gray text.

| Ratio | Method | PPL ↓ | | Accuracy ↑ | | | | | | | |
|---|---|---|---|---|---|---|---|---|---|---|---|
| | | Wiki | C4 | ARC-e | ARC-c | PIQA | Openb. | WinoG. | MathQA | HellaS. | Avg |
| 1.0 | Original | 5.68 | 7.34 | 0.76 | 0.38 | 0.79 | 0.34 | 0.70 | 0.27 | 0.51 | 0.64 |
| 0.8 | Basis Sharing | 7.74 | 15.03 | 0.66 | **0.36** | 0.71 | **0.28** | 0.66 | **0.25** | 0.46 | 0.56 |
| | Swift-SVD | **7.71** | **11.27** | **0.67** | 0.35 | **0.73** | **0.28** | **0.68** | 0.24 | **0.47** | **0.57** |
| 0.6 | Basis Sharing | **12.39** | 41.28 | 0.52 | **0.27** | 0.62 | 0.22 | 0.61 | **0.23** | 0.35 | 0.47 |
| | Swift-SVD | 12.69 | **22.10** | **0.53** | 0.25 | **0.63** | **0.23** | **0.63** | **0.23** | **0.36** | **0.48** |

For uniform compression, Swift-SVD outperforms SVD-LLM(W) and Dobi-SVD(w/o) at both compression ratios. For dynamic compression, Dobi-SVD(w) uses the training-based dynamic rank allocation from Dobi-SVD's original paper; Swift-SVD(C) follows the pure layer-wise loss-based strategy of SVD-LLM v2; Swift-SVD* jointly considers layer-wise loss and layer importance. Swift-SVD* consistently achieves the highest accuracy, and both Swift-SVD variants substantially outperform Dobi-SVD(w), which suffers from its training-based allocation on this architecture.

*Table B.3.* Zero-shot accuracy on Qwen3-4B. Baseline results are reported from their original papers. The Original row is highlighted with gray text.

| Ratio | Method | ARC-e | PIQA | Openb. | WinoG. | HellaS. | MathQA | Avg |
|---|---|---|---|---|---|---|---|---|
| 1.0 | Original | 0.81 | 0.75 | 0.30 | 0.66 | 0.52 | 0.40 | 0.57 |
| 0.8 | SVD-LLM(W) | 0.64 | 0.62 | 0.24 | 0.57 | 0.40 | 0.27 | 0.46 |
| | Dobi-SVD(w/o) | 0.62 | 0.61 | 0.23 | 0.55 | 0.39 | 0.26 | 0.44 |
| | Dobi-SVD(w) | 0.31 | 0.60 | 0.19 | 0.55 | 0.30 | 0.21 | 0.36 |
| | Swift-SVD | 0.69 | 0.70 | 0.26 | 0.61 | 0.42 | 0.29 | 0.50 |
| | Swift-SVD(C) | 0.69 | 0.69 | 0.26 | 0.60 | 0.41 | 0.27 | 0.49 |
| | **Swift-SVD*** | **0.71** | **0.72** | **0.29** | **0.63** | **0.46** | **0.30** | **0.52** |
| 0.6 | SVD-LLM(W) | 0.41 | 0.56 | 0.18 | 0.43 | 0.29 | 0.19 | 0.34 |
| | Dobi-SVD(w/o) | 0.40 | 0.54 | 0.17 | 0.39 | 0.28 | 0.18 | 0.33 |
| | Dobi-SVD(w) | 0.27 | 0.53 | 0.16 | 0.53 | 0.25 | 0.18 | 0.32 |
| | Swift-SVD | 0.54 | 0.65 | 0.23 | 0.59 | 0.35 | 0.23 | 0.43 |
| | Swift-SVD(C) | 0.54 | 0.65 | 0.20 | 0.59 | 0.33 | 0.20 | 0.42 |
| | **Swift-SVD*** | **0.57** | **0.67** | **0.23** | **0.60** | **0.38** | **0.24** | **0.45** |

## B.4. Scalability to Larger Models

We evaluate Swift-SVD's scalability on Qwen-32B, a model significantly larger than those tested in the main text. All experiments are conducted on a single H800 (80GB) GPU. Under this hardware constraint, activation-aware baselines such as SVD-LLM and Dobi-SVD fail with OOM errors when compressing Qwen-32B, while Swift-SVD completes successfully—highlighting its memory efficiency advantage at scale. We therefore compare against naive SVD (direct low-rank decomposition without activation awareness) as a baseline. Sundrani (2025) do not release source code, precluding direct comparison.

Table B.4 reports perplexity and zero-shot accuracy results. Swift-SVD consistently outperforms naive SVD by a large margin across all compression ratios, demonstrating the importance of activation awareness at the 32B scale. Notably, at compression ratio 0.8, Swift-SVD improves average accuracy from 0.42 (uncompressed) to 0.59. This improvement under moderate compression is not observed in smaller (≤8B) models and may reflect a mild regularization effect in more overparameterized models, consistent with prior observations (Sundrani, 2025).

## B.5. Singular Value Distribution

As visualized in Figure B.1, the singular value distribution of both Key and Value modules exhibits a pronounced spectral disparity. Even on a logarithmic scale, we observe a massive magnitude gap: the dominant singular values reach up to $10^5$, while the median values hover significantly lower. Crucially, this extreme unevenness renders rank allocation methods based solely on Frobenius norm minimization ineffective. Since the Frobenius loss is disproportionately sensitive to large singular

*Table B.4.* Perplexity and zero-shot accuracy of Swift-SVD on Qwen-32B (single H800, 80GB). Naive SVD denotes direct low-rank decomposition without activation awareness. PPL results for Naive SVD are omitted as they are too large to be informative. The Original row is highlighted with gray text.

| Ratio | Method | PPL ↓ | | Accuracy ↑ | | | | | | | |
|---|---|---|---|---|---|---|---|---|---|---|---|
| | | Wiki | C4 | ARC-e | ARC-c | PIQA | Openb. | WinoG. | MathQA | HellaS. | Avg |
| 1.0 | Original | 7.60 | 12.40 | 0.25 | 0.23 | 0.50 | 0.36 | 0.73 | 0.21 | 0.64 | 0.42 |
| 0.8 | Naive SVD | — | — | 0.27 | 0.23 | 0.60 | 0.21 | 0.51 | 0.21 | 0.30 | 0.33 |
| | Swift-SVD | **9.38** | **15.45** | **0.79** | **0.49** | **0.78** | **0.36** | **0.73** | **0.43** | **0.58** | **0.59** |
| 0.6 | Naive SVD | — | — | 0.25 | 0.22 | 0.55 | 0.16 | 0.50 | 0.19 | 0.26 | 0.30 |
| | Swift-SVD | **12.59** | **22.99** | **0.73** | **0.42** | **0.74** | **0.29** | **0.72** | **0.30** | **0.49** | **0.53** |
| 0.4 | Naive SVD | — | — | 0.24 | 0.20 | 0.52 | 0.15 | 0.41 | 0.18 | 0.25 | 0.28 |
| | Swift-SVD | **23.90** | **56.79** | **0.52** | **0.26** | **0.67** | **0.21** | **0.64** | **0.23** | **0.37** | **0.41** |

values, such methods suffer from severe numerical bias, often assigning insufficient ranks to layers with smaller spectral norms. Furthermore, our empirical analysis reveals a high negative correlation between layer-wise compressibility and importance. These observations motivate our proposed dynamic allocation strategy.

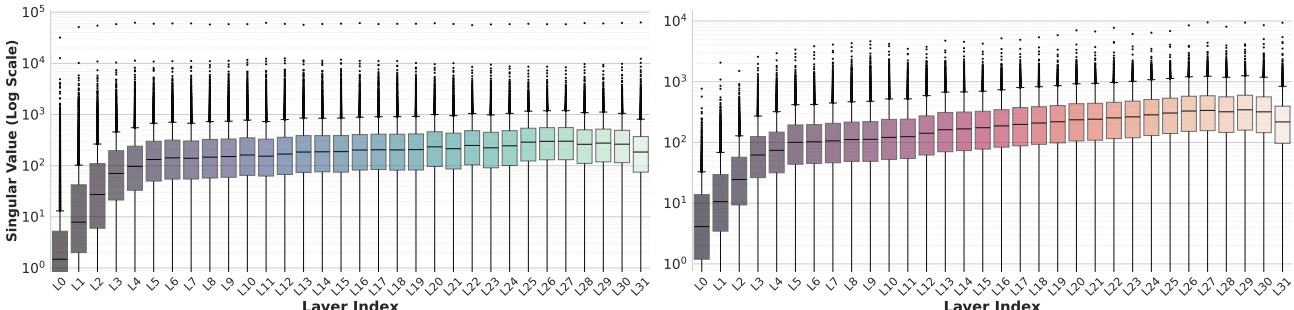

*Figure B.1.* Singular value distribution of Key (left) and Value (right) modules in Llama-7B. The y-axis is presented on a logarithmic scale to visualize the magnitude differences across layers.

## B.6. Throughput

We further evaluate the system performance across varying batch sizes as shown in Figure B.2. The experimental results demonstrate that as the compression ratio increases, our method enhances inference efficiency by simultaneously boosting throughput and alleviating HBM pressure.

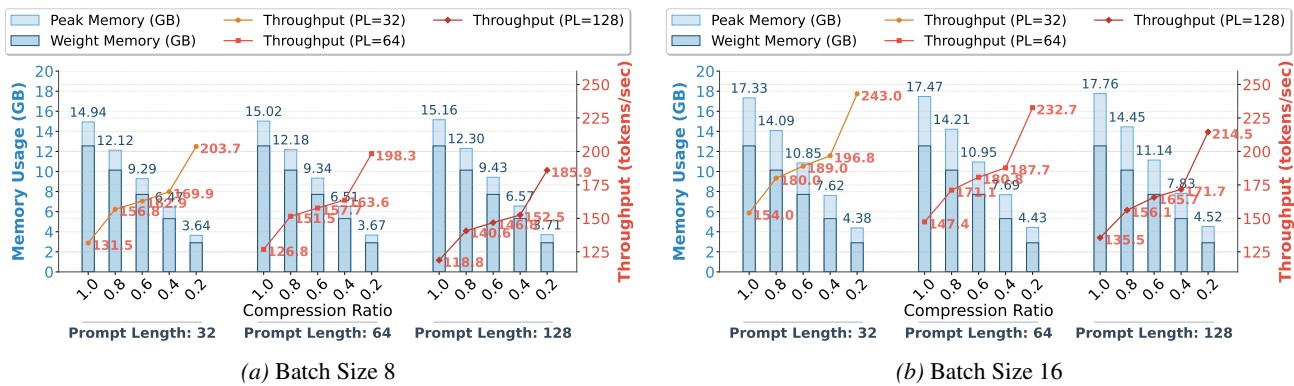

*Figure B.2.* Throughput improvement and memory efficiency under different batch sizes. The generated sequence length is 1024.

## B.7. Serving Throughput Comparison Across Methods

We compare serving throughput (tokens/sec) across SVD-based compression methods under both uniform and dynamic rank allocation on LLaMA-2-7B, conducted on an H800 GPU with batch size 16, prompt length 32, and output token length 1024. **SVD-LLM(W)**, **Dobi-SVD(w/o)**, and **Swift-SVD** denote uniform compression of the corresponding methods. **Swift-SVD\*** denotes dynamic compression that jointly considers layer-wise loss and layer importance (our full method). **Swift-SVD(C)** follows the pure loss-based dynamic strategy of SVD-LLM V2. **Dobi-SVD(w)** uses the training-based rank allocation from Dobi-SVD's official repository. Results are shown in Table B.5. Three observations emerge.

**Uniform compression.** SVD-LLM(W), Dobi-SVD(w/o), and Swift-SVD achieve nearly identical throughput at the same compression ratio (left block of Table B.5), confirming that equal compression ratio implies equal throughput under uniform rank allocation, where all layers share the same rank and thus identical GEMM dimensions.

**Dynamic compression — Swift-SVD.** Swift-SVD\* and Swift-SVD(C) retain throughput close to their respective uniform baselines (right block of Table B.5), with no significant degradation observed. One possible explanation is that, to achieve good compression quality, dynamic allocation tends not to deviate dramatically from uniform ranks, so the variation in GEMM dimensions may not be large enough to cause observable throughput differences in practice.

**Dynamic compression — Dobi-SVD(w).** Dobi-SVD(w) exhibits substantially lower throughput than its uniform counterpart (right block of Table B.5). We attribute this to its *inter-type* rank allocation, which may assign higher ranks to k_proj and v_proj at the expense of other projection types, enlarging KV cache dimensions and increasing memory traffic per decoding step—directly degrading throughput in the memory-bandwidth-bound decoding phase. Table B.6 shows the per-type average ranks from Dobi-SVD(w)'s allocation. For example, at ratio 0.8 the uniform rank for Q/K/V is 1638, yet the mean ranks of k_proj (1661) and v_proj (1743) both exceed this baseline. To validate this hypothesis, we compress LLaMA-2-7B using the per-type average ranks from Dobi-SVD(w)'s allocation and measure the resulting throughput (reported in Table B.6), which closely matches Dobi-SVD(w) itself (Table B.5: 104.4 / 117.5 / 131.6 tokens/sec at ratios 0.8/0.6/0.4).

*Table B.5.* Serving throughput (tokens/sec) on LLaMA-2-7B (H800) under uniform and dynamic rank allocation.

| Ratio | Uniform Compression | | | Dynamic Compression | | |
|---|---|---|---|---|---|---|
| | Swift-SVD | SVD-LLM(W) | Dobi-SVD(w/o) | Swift-SVD\* | Swift-SVD(C) | Dobi-SVD(w) |
| 0.8 | 192.9 | 192.7 | 193.3 | 193.7 | 191.4 | 104.4 |
| 0.6 | 209.2 | 208.1 | 207.8 | 207.3 | 206.3 | 117.5 |
| 0.4 | 216.8 | 216.5 | 216.8 | 216.0 | 215.4 | 131.6 |

*Table B.6.* Per-type average ranks from Dobi-SVD(w)'s dynamic allocation on LLaMA-2-7B, alongside the uniform rank baseline. The Throughput column reports serving throughput when compressing with these per-type average ranks, confirming that the enlarged k_proj and v_proj ranks are the primary cause of throughput degradation.

| Ratio | Uniform rank | q_proj | k_proj | v_proj | o_proj | gate_proj | up_proj | down_proj | Throughput |
|---|---|---|---|---|---|---|---|---|---|
| 0.8 | 1638 | 1627 | 1661 | 1743 | 1588 | 2365 | 2379 | 2385 | 101.3 |
| 0.6 | 1229 | 1213 | 1230 | 1294 | 1189 | 1773 | 1781 | 1812 | 113.4 |
| 0.4 | 819 | 794 | 775 | 839 | 773 | 1207 | 1206 | 1227 | 130.8 |

## B.8. Throughput on Longer Generation Tasks

A potential concern with KV cache compression is that up-projecting all $L$ cached KV latents at each decoding step may introduce significant overhead as sequence length grows. We address this by noting that the up-projection matrix has fixed dimensions, so the additional computational cost per cached token is constant—the total overhead per decoding step scales as $\mathcal{O}(L)$, not $\mathcal{O}(L^2)$. Crucially, the memory bandwidth savings from reading the compressed KV cache also scale as $\mathcal{O}(L)$, so the throughput gain of the compressed model is preserved at longer sequence lengths.

To verify this empirically, we measure throughput on Mistral-7B at generation lengths of 1024 and 8192 tokens on an H800 GPU (batch size 16, prompt length 32). Longer sequences exceed the memory capacity of RTX 5090 due to the large total KV cache size, which is why we use H800 here. As shown in Table B.7, token throughput decreases as sequence length grows—an inherent property of autoregressive decoding, where longer generation requires reading increasingly large KV caches during attention computation, amplifying the memory-bandwidth bottleneck. This behavior is consistent across both

compressed and uncompressed models. Importantly, the throughput gain at length 8192 is comparable to or exceeds that at length 1024 for each compression ratio, demonstrating that Swift-SVD remains practically efficient for longer generation tasks.

*Table B.7.* Throughput (tokens/sec) and gain over uncompressed baseline on Mistral-7B (H800, bs=16, prompt length 32) at generation lengths 1024 and 8192.

|  | **Length 1024** | | **Length 8192** | |
| --- | --- | --- | --- | --- |
| Ratio | Throughput | Gain | Throughput | Gain |
| 1.0 | 438.86 | — | 140.39 | — |
| 0.8 | 457.31 | +4% | 144.98 | +3% |
| 0.6 | 473.38 | +8% | 169.17 | +20% |
| 0.4 | 497.29 | +12% | 181.23 | +29% |
| 0.2 | 551.04 | +26% | 187.45 | +34% |

## B.9. Layer-wise NER and Importance Results

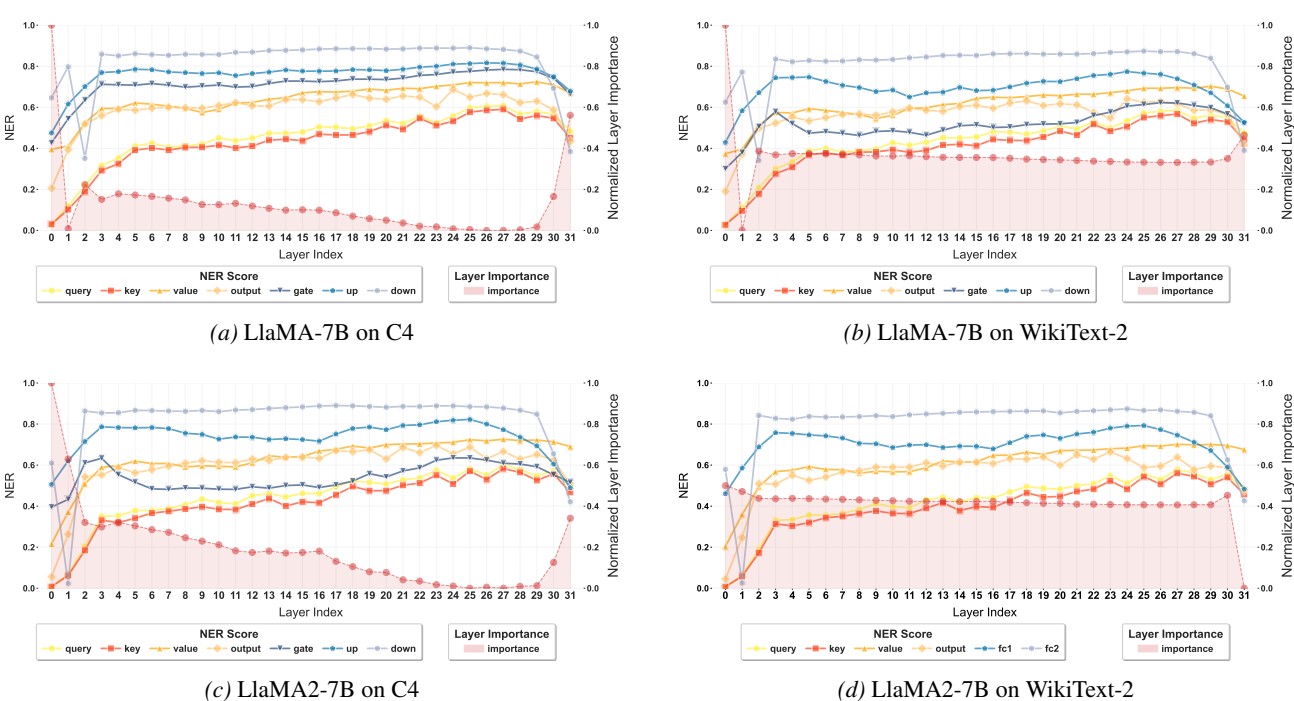

*(a)* LlaMA-7B on C4

*(b)* LlaMA-7B on WikiText-2

*(c)* LlaMA2-7B on C4

*(d)* LlaMA2-7B on WikiText-2

*Figure B.3.* Layer-wise NER and Importance comparison (Part I).

## B.10. Contents Generated From Compressed LLMs

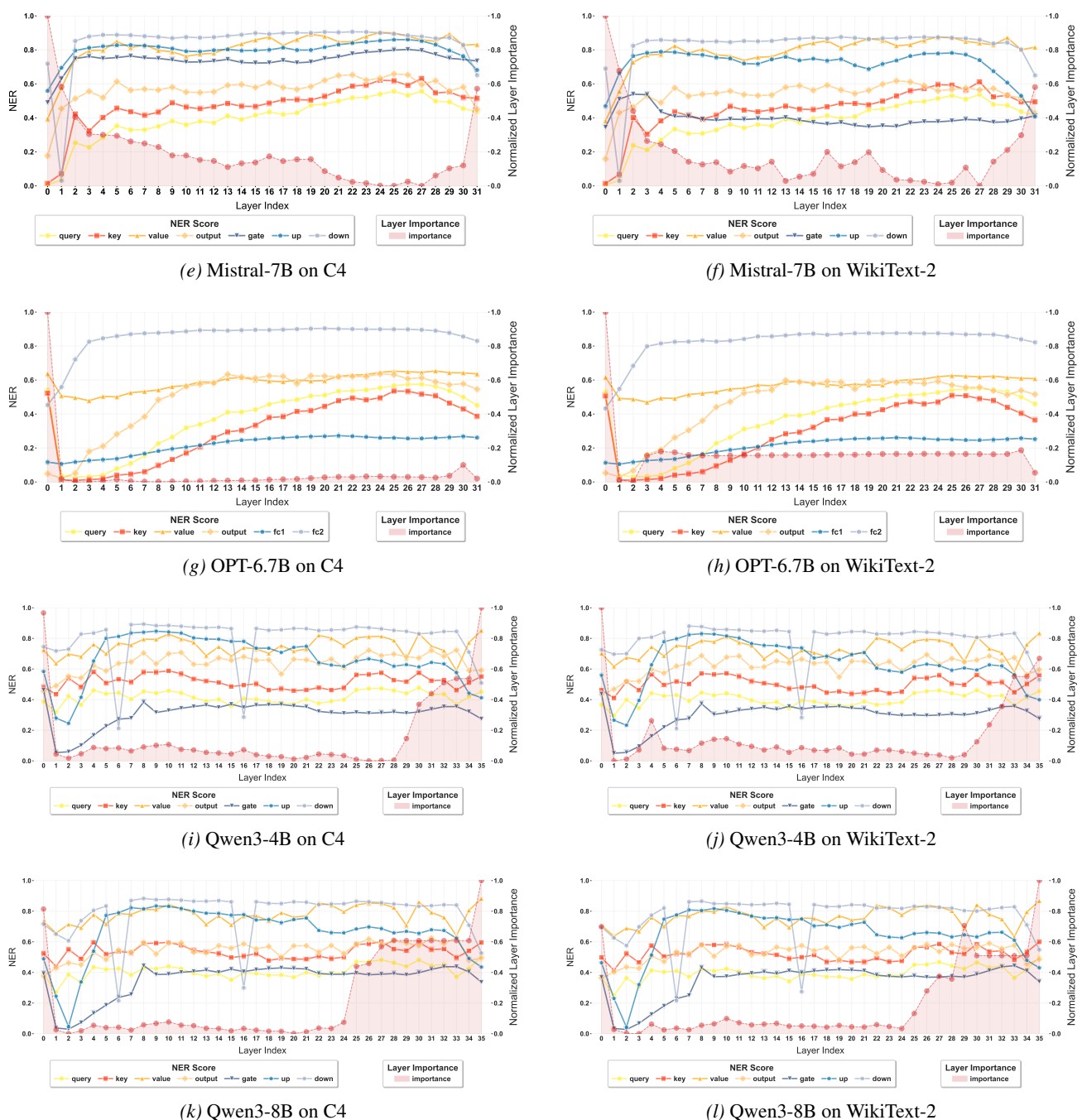

*Figure B.3.* Layer-wise NER and Importance comparison (Part II).

| RATIO | WikiText-2 | C4 |
|---|---|---|
| Original | **What color is the sky on a clear day?** On a clear day, the sky appears blue due to the way sunlight scatters through the atmosphere, a phenomenon known as Rayleigh scattering. | |
| 0.8 | **What color is the sky on a clear day?** Blue, green or white. The answer to this question should be blue. Because of the atmosphere and sunlight, it's blue. However some people might think that it's green because they are looking at trees in front of the sky. Others may say that it's white due to clouds. But actually it's blue. The same way as with other colors like red or yellow, the color of the sky can change depending on weather conditions. | **What color is the sky on a clear day?** The sky is blue on a clear day. The sky appears blue because sunlight is scattered by the atmosphere. |
| 0.6 | **What color is the sky on a clear day?** Blue or white. () in what colors are to be used for the sky. The sun and moon have their own light, but it's not that they can change into red or green, which appears as dark clouds when we observe them from high altitude. In this case of being able to see all skies, these colors would be so different with such great brightness. For example , you could imagine that some people might find something similar to the sky's color. I think perhaps the sky is blue." | **What color is the sky on a clear day?** The answer is blue. But this question might be confusing because it's not possible to see what colors are in the atmosphere when you look at it from Earth. So I need to find out what happens if there's no clouds, and then how much light would reflect into the atmosphere? If we have some cloud cover over the earth and only sunlight comes through, that could make an effect of reflection by something like reflected sunlight or light being absorbed. |
| 0.4 | **What color is the sky on a clear day?** It has blue colors. The sky's white coloring in black, with yellow hues of purple that are gray. It s colored as red, and its colors have green tones which are dark. This landscape area is brown to some areas where they are not covered by the natural vegetation but they contain many small plants such trees and other species like plant wood. With these trees having a forest structure from their tree and being surrounded by various species including woodland. | **What color is the sky on a clear day?** Options: - A) white, b ) green , c ) blue , d ) yellow. The answer will be blue. |
| Original | **What is the responsibility of the AI assistant?** The responsibility of an AI assistant primarily lies in providing accurate, helpful, and ethical assistance to users. This includes adhering to guidelines, maintaining privacy, avoiding harmful or biased content, and ensuring that the information provided is reliable and up-to-date. Additionally, AI assistants should respect user autonomy, offer transparency about their capabilities and limitations, and avoid any form of deception or manipulation. | |
| 0.8 | **What is the responsibility of the AI assistant?** First, to ensure that its responses are accurate and reliable, in accordance with human standards. The second task of AI assistants is to be a helpful companion for humans. This means they can help people by providing information about their own lives, or helping them solve problems. In addition, AI assistance can serve as an aid to enhance intelligence through learning from data collected by other AI systems. And finally, AI assistants may also have responsibilities related to ethical issues such as privacy protection. | **What is the responsibility of the AI assistant?** The AI assistant is responsible for providing accurate information and helping users with their questions. |
| 0.6 | **What is the responsibility of the AI assistant?** "In a way that would be useful for human beings. The AI assistant can help humans to find out how they are not making their own decisions, which may lead them to confusion and make errors. That is also helpful when people try to create new ideas or thoughts. It's possible that these actions could have been used by other people who want to take advantage with this assistance from AI assistants." | **What is the responsibility of the AI assistant?** The ai assistant must do all tasks required by people. In order to provide a good solution, they need to be responsible for what happens and can have an accurate response when it's needed. Answer: The Ai assistant should perform tasks that require human attention. It has responsibility for doing tasks that require humans' attention so as they will get proper answers with accuracy. |
| 0.4 | **What is the responsibility of the AI assistant?** The AI assistant serves as one agent who may serve both or roles within organizational activities, but not being affiliated to any organization. | **What is the responsibility of the AI assistant?** The user has a total number of tasks, and they can be assigned with some tools. |

*Table B.8.* Comparison of AI assistant responses across different datasets and compression ratios.

