# OpenReview forum: "Swift-SVD: Theoretical Optimality Meets Practical Efficiency in Low-Rank LLM Compression"
_ICML.cc/2026/Conference — ICML 2026 regular_

### Official Review · Reviewer_Un8K · 2026-03-07

**Soundness:** 3
**Presentation:** 3
**Significance:** 2
**Originality:** 2
**Overall Recommendation:** 4
**Confidence:** 4

**Summary:**

This paper proposes Swift-SVD, a training-free, activation-aware low-rank compression framework for large language models. The method formulates activation-aware weight compression as a closed-form spectral optimization problem, showing that the optimal compressed weight can be derived from the singular vectors of the activation–weight product. Based on this theoretical result, Swift-SVD performs efficient eigenvalue decomposition to obtain optimal low-rank approximations and introduces a dynamic rank allocation strategy that considers both layer-wise reconstruction loss and layer importance. The approach avoids the numerical instability and computational overhead of prior SVD-based compression methods such as SVD-LLM and Dobi-SVD. Experiments demonstrate that Swift-SVD achieves near-theoretical minimal reconstruction error while improving compression efficiency and inference throughput across multiple models and compression ratios.

**Compliance With Llm Reviewing Policy:**

Affirmed.

**Final Justification:**

The rebuttal from the authors address all my concerns.

**Key Questions For Authors:**

1. Could the authors provide comparisons with baselines such as SVD-LLM, Dobi-SVD, or MoDeGPT on Qwen3 to better contextualize the performance improvements?
2. Could the authors clarify why the proposed formulation eliminates the need for fine-tuning? For example, is there theoretical or empirical evidence showing that the reconstruction error or approximation quality is sufficient to maintain model performance without post-compression training?
3. Could the authors provide comparisons or analysis showing how the proposed rank allocation strategy differs from or improves upon those used in prior methods?
4. Could the authors clarify what aspects of Swift-SVD are fundamentally new compared with prior activation-aware SVD-based compression methods?

**Limitations:**

Yes

**Strengths And Weaknesses:**

## Strengths
1. **Training-free and numerically stable compression framework.**
Swift-SVD avoids expensive gradient-based optimization and calibration-heavy procedures by computing the optimal solution through eigenvalue decomposition of activation statistics. Compared with prior methods such as SVD-LLM and Dobi-SVD, the approach improves numerical stability and computational efficiency, reducing implementation complexity while maintaining theoretical optimality.
2. **Evaluation on modern LLM architectures.**
The authors evaluate Swift-SVD on newer model architectures, including Qwen3, demonstrating that the proposed method generalizes beyond earlier transformer models and remains effective on more recent LLMs.

## Weaknesses
1. **Incomplete baseline comparison on newer models.**
Although the authors apply the proposed method to Qwen3, they do not report comparisons with relevant baseline methods such as SVD-LLM, Dobi-SVD, or MoDeGPT on this model. Without such comparisons, it is difficult to assess whether the proposed method provides meaningful accuracy improvements over existing approaches on modern architectures.
2. **Insufficient explanation of the fine-tuning-free property.**
The paper claims that Swift-SVD does not require additional fine-tuning, whereas prior methods such as SVD-LLM and Dobi-SVD typically rely on post-compression fine-tuning to recover model performance. However, the paper does not clearly explain why the proposed formulation eliminates the need for fine-tuning. Additional explanation or theoretical analysis—such as reconstruction error guarantees or tighter error bounds—would help clarify the advantages of the proposed method.
3. **Limited novelty compared with recent methods.**
The novelty of the proposed approach appears somewhat limited. Techniques such as activation-aware (or scaled) SVD and dynamic rank allocation have been explored in recent work, including MoDeGPT and SVD-LLMv2. However, the paper does not provide comparisons between the proposed rank allocation strategy and those used in prior methods, making it difficult to evaluate the methodological contribution.

---

> ### Author Rebuttal · Authors · 2026-03-31
>
> **W1&Q1**: Could the authors provide comparisons with baselines on Qwen3?
>
> **A**: We compared Swift-SVD with SVD-LLM and Dobi-SVD on Qwen3. MoDeGPT is excluded as it does not natively support the Qwen3 architecture, and adapting it would require non-trivial modifications beyond the scope of this rebuttal. As shown in the table below, Swift-SVD achieves higher average accuracy than both SVD-LLM and Dobi-SVD across compression ratios.
>
> ### Table: Accuracy
> | Method | ratio | ARC-e | PIQA | OpenB | WinoG | HellaSwag | MathQA | Avg |
> |--------|------|-------|------|-------|-------|-----------|--------|------|
> | Original | 1.0 | 0.81 | 0.75 | 0.30 | 0.66 | 0.52 | 0.40 | 0.57 |
> | SVD-LLM | 0.8 | 0.64 | 0.62 | 0.24 | 0.57 | 0.40 | 0.27 | 0.46 |
> | SVD-LLM | 0.6 | 0.41 | 0.56 | 0.18 | 0.43 | 0.29 | 0.19 | 0.34 |
> | Dobi-SVD | 0.8 | 0.62 | 0.61 | 0.23 | 0.55 | 0.39 | 0.26 | 0.44 |
> | Dobi-SVD | 0.6 | 0.40 | 0.54 | 0.17 | 0.39 | 0.28 | 0.18 | 0.33 |
> | Ours | 0.8 | 0.69 | 0.70 | 0.26 | 0.61 | 0.42 | 0.29 | 0.50 |
> | Ours | 0.6 | 0.54 | 0.65 | 0.23 | 0.59 | 0.35 | 0.23 | 0.43 |
> ---
> **W2&Q2**: Could the authors clarify why the proposed formulation eliminates the need for fine-tuning?
>
> **A**:  We thank the reviewer for this question. Even though Swift-SVD focuses on training-free low-rank compression, it is compatible with post-compression fine-tuning. The compressed model produced by Swift-SVD can serve directly as a LoRA initialization, following the same paradigm as SVD-LLM.
>
> The quality of initialization can affect how well LoRA fine-tuning recovers model performance. As reported in SVD-LLM's paper, at compression ratio 0.4 (5.3GB) on LLaMA-7B, SVD-LLM's compressed model achieves only 0.11 average accuracy, and even after LoRA fine-tuning, recovers to only 0.30--still below Swift-SVD's 0.34 achieved by training-free compression alone (Table 1). This suggests that a poor initialization can limit the effectiveness of LoRA fine-tuning. Since Swift-SVD consistently provides a stronger initialization, we expect combining it with LoRA to yield further gains, and leave this as future work.
>
> ---
>
> **W3&Q3**: Could the authors provide comparisons or analysis showing how the proposed rank allocation strategy differs from or improves upon those used in prior methods?
>
> **A**: We thank the reviewer for this question. Our rank allocation differs from prior methods in both allocation strategy and performance.
>
> **(1) Difference in rank allocation**. Swift-SVD jointly considers layer-wise loss and layer importance, and performs grid search over the trade-off between them.  In contrast, SVD-LLM V2 allocates ranks purely based on layer-wise loss, and must recompute W*_k and evaluate \epsilon*_k=||XW − XW*_k|| for each candidate rank, making exhaustive search prohibitively expensive. Dobi-SVD relies on differentiable training to allocate rank dynamically, which requires gradient-based optimization.
>
> **(2) Performance comparison**. As shown in Table 6, pure layer-wise loss-based allocation Swift-SVD(C)--implemented following the strategy proposed in SVD-LLM V2--underperforms even uniform compression, confirming that layer-wise loss alone is insufficient. Further, as shown in Table 1, Swift-SVD* consistently outperforms Dobi-SVD(w)--the dynamic rank allocation variant of Dobi-SVD-- across models and datasets. Moreover, at compression ratio 0.4 on C4, Dobi-SVD(w) performs worse than its uniform counterpart Dobi-SVD(w/o), suggesting that training-based rank allocation may converge to a suboptimal solution in some cases.
>
> ---
>
> **W3&Q4**: Could the authors clarify what aspects of Swift-SVD are fundamentally new compared with prior activation-aware SVD-based compression methods?
>
> **A**: We thank the reviewer for this question. Swift-SVD shares the same activation-aware objective as prior works, including ASVD, SVD-LLM, SVD-LLM V2, Dobi-SVD. Our contribution is not a new objective, but a new close-form solution along with a practically efficient implementation. We summarize the key distinctions below.
>
> **New closed-form results**. Swift-SVD shows that the eigendecomposition of Y^TY gives V and \sigma^2, and the closed form optimal solution is W*_k=W V_k V_k^T, and \epsilon*_k = \sqrt{\sum{/sigma^2}}. SVD-LLM also derives both, but in a form that requires Cholesky decomposition of XX^T, an additional SVD, and matrix inversion--operations that introduce numerical instability. SVD-LLM V2 derives an different form of optimal W*_k but provides no closed-form \epsilon*_k. Dobi-SVD derives W*_k as well but obtains V via PCA over Y, and the closed-form of \epsilon*_k is not proved.
>
> **Computational efficiency**. Swift-SVD reduces the problem to a single eigendecomposition of Y^TY, which can be accumulated incrementally as C = Σ Y_i^T Y_i. In contrast, SVD-LLM requires Cholesky + SVD + matrix inversion, SVD-LLM V2 requires two successive SVDs, and Dobi-SVD proposed IPCA, which requires O(S) SVD operations where S is the number of calibration samples.

---

> > ### Author Rebuttal · Reviewer_Un8K · 2026-04-04
> >
> > We thank the authors for their rebuttal, which partially addresses my concerns.
> >
> > I am confused by the authors’ response to W2 and Q2. Are the reported accuracy results obtained with or without additional training? Given that methods such as SVD-LLM and Dobi-SVD typically require fine-tuning to achieve their best performance, it would be helpful if the authors could clarify what setting the results in Table 1 correspond to.
> >
> > Additionally, could the authors elaborate on how the proposed rank allocation differs from the Block Influence (BI) scores used in [1, 2]?
> >
> > [1] Shortgpt: Layers in large language models are more redundant than you expect
> >
> > [2] Modegpt: Modular decomposition for large language model compression

---

> > > ### Author Response · Authors · 2026-04-07
> > >
> > > As the rebuttal period concludes, we sincerely thank you for your thoughtful and constructive feedback. We believe the additional experimental results have addressed the main concerns and improved the overall quality of the paper. We kindly hope you would consider updating your evaluation.
> > >
> > > Q: I am confused by the authors’ response to W2 and Q2. Are the reported accuracy results obtained with or without additional training? Given that methods such as SVD-LLM and Dobi-SVD typically require fine-tuning to achieve their best performance, it would be helpful if the authors could clarify what setting the results in Table 1 correspond to.
> > >
> > > A: We thank the reviewer for seeking this clarification. All results in Table 1 are training-free and obtained under uniform rank allocation. No fine-tuning or post-compression training is applied to any method. Each method compresses each layer to the same rank k and obtains the optimal low-rank approximation W*_k via its own procedure: SVD-LLM uses Cholesky decomposition of $XX^T$ followed by an additional SVD, Dobi-SVD uses IPCA, and Swift-SVD uses a single eigendecomposition of $Y^TY.$ We will clarify this in the caption of Table 1 in the next version.
> > >
> > > To further elaborate the experimental analysis, we compare methods under dynamic rank allocation on Qwen3-4B. The results are shown in Table 2 below, where Dobi-SVD(w) uses the training-based dynamic rank allocation proposed by Dobi-SVD (following the training settings reported in their paper, with optimal W*_k obtained via IPCA), Swift-SVD* jointly considers layer-wise loss and layer importance, and Swift-SVD(C) follows the pure layer-wise loss-based strategy of SVD-LLM V2.
> > >
> > > As shown in Table 2, Swift-SVD* consistently outperforms Swift-SVD(C) across both compression ratios, confirming the benefit of jointly considering layer importance alongside layer-wise reconstruction loss. At 0.8 compression ratio, Swift-SVD* achieves 0.52 average accuracy, outperforming Swift-SVD(C) (0.49) and substantially outperforming Dobi-SVD(w) (0.36). At 0.6 compression ratio, Swift-SVD* achieves 0.45 vs. Swift-SVD(C)'s 0.42 and Dobi-SVD(w)'s 0.32, demonstrating consistent advantages across compression levels.
> > >
> > > **Table 2: Accuracy of Qwen3-4B under dynamic rank allocation**
> > > | Method | ratio | ARC-e | PIQA | OpenB | WinoG | HellaSwag | MathQA | Avg |
> > > |--------|------|-------|------|-------|-------|-----------|--------|------|
> > > | Original | 1.0 | 0.81 | 0.75 | 0.30 | 0.66 | 0.52 | 0.40 | 0.57 |
> > > | Dobi-SVD(w) | 0.8 | 0.31 | 0.60 | 0.19 | 0.55 | 0.30 | 0.21 | 0.36|
> > > | Dobi-SVD(w) | 0.6 | 0.27 | 0.53 | 0.16 | 0.53 | 0.25 | 0.18 | 0.32 |
> > > | Swift-SVD(C)| 0.8 | 0.69 | 0.69 | 0.26 | 0.60 | 0.41 | 0.27 | 0.49 |
> > > | Swift-SVD(C)| 0.6 | 0.54 | 0.65 | 0.20 | 0.59 | 0.33 | 0.20 | 0.42 |
> > > | Swift-SVD*| 0.8 | 0.71 | 0.72 | 0.29 | 0.63 | 0.46 | 0.30 | 0.52 |
> > > | Swift-SVD*| 0.6 | 0.57 | 0.67 | 0.23 | 0.60 | 0.38 | 0.24 | 0.45|
> > >
> > > Q: Additionally, could the authors elaborate on how the proposed rank allocation differs from the Block Influence (BI) scores used in [1, 2]?
> > >
> > > [1] Shortgpt: Layers in large language models are more redundant than you expect
> > >
> > > [2] Modegpt: Modular decomposition for large language model compression
> > >
> > > A: We thank the reviewer for pointing out these related works. We clarify the key differences in rank allocation below and will add both to the related work section in the next version.
> > >
> > > ShortGPT uses BI scores to identify and remove entire redundant layers, whereas Swift-SVD retains all layers and uses layer importance jointly with layer-wise reconstruction loss to allocate different compression ranks across layers.
> > >
> > > MoDeGPT uses BI scores for per-layer rank allocation. The BI score, like our layer importance metric, reflects each layer's contribution to end-to-end model performance. However, MoDeGPT relies on BI score alone, while Swift-SVD jointly considers both layer importance and layer-wise reconstruction loss--capturing both end-to-end and local compression effects. We believe this joint consideration is the key to effective rank allocation: as shown in our ablation (Table 6), Swift-SVD(I)--which uses layer importance alone for rank allocation, similar in spirit to MoDeGPT--underperforms uniform compression. We note that MoDeGPT addresses a different compression problem (modular decomposition at the block level) from Swift-SVD (per-matrix low-rank decomposition into W = AB), which may explain why BI-only allocation is sufficient for MoDeGPT but not for Swift-SVD.

---

### Official Review · Reviewer_9ZPL · 2026-03-09

**Soundness:** 3
**Presentation:** 3
**Significance:** 3
**Originality:** 3
**Overall Recommendation:** 4
**Confidence:** 3

**Summary:**

This paper proposes Swift-SVD, a training-free method for low-rank LLM compression that uses activation statistics to reduce both weight and KV-cache memory. It derives a closed-form solution using one eigendecomposition, then adds dynamic rank allocation across layers. Experiments report better accuracy-efficiency trade-offs and faster compression than prior SVD-based methods.

**Compliance With Llm Reviewing Policy:**

Affirmed.

**Final Justification:**

Regarding Weakness 1, the rebuttal provides a clearer empirical motivation and explains why exhaustive search becomes tractable in Swift-SVD. However, the proposed score and allocation rule remain largely heuristic rather than theoretically justified. In particular, the specific form of the compressibility score in Equation 12 is not rigorously derived from a loss function, but more like an ad-hoc formulation.

Regarding Limitation 1, my concern is only partially addressed. The method itself still appears to have the best accuracy at the relatively mild compression ratio of 0.8. The accuracy drops quickly at 0.6 or 0.4 compared to that of 0.8, so the practical impact under more aggressive compression remains limited.

Other concerns are mostly resolved. Overall, I maintain my positive score.

**Key Questions For Authors:**

The paper says in Section 4.1 that "all evaluations are conducted under identical experimental protocols", but Table 1 caption also says "baseline results are reported from their original papers". Can the authors clarify if all methods are evaluated under exactly the same calibration data and other relevant settings?

**Limitations:**

1. Near-original performance is only obtained at mild compression ratios, which may limit practical impact. Table 1 suggests that only the models compressed at the 0.8 ratio retain accuracy close to the uncompressed models. In contrast, other compression approaches, such as quantization, often preserve near-original accuracy at substantially lower compression ratios.

2. The paper shows results mainly on moderate-size (<= 8B) models, so scalability remains somewhat unclear.

**Strengths And Weaknesses:**

**Strengths**

1. The paper tackles a practical problem of compressing both LLM weights and KV cache for cheaper inference. This makes the contribution relevant to real deployment settings.

2. The method is simple and well motivated. It gives a closed-form activation-aware solution using a single eigendecomposition.

3. The dynamic rank allocation is a sensible extension beyond uniform compression. It uses both layer-wise loss and layer importance.

4. The experiments are fairly broad and report gains over prior SVD-based baselines. The paper also supports its claims with efficiency and numerical stability results.

---

**Weaknesses**

1. The dynamic rank allocation strategy appears heuristic. The compressibility score (Equation 12), the rank allocation strategy (Algorithm 2), and grid search are empirically motivated, but the paper does not provide a strong theoretical justification for why this particular non-trivial formulation should be preferred over other potential alternatives.

2. Minor presentation issues. Figure 3 misses the label for y axis. Table 4 needs to specify the GPU type. Section 3.2.2 Line 253 "retention ration" should be "retention ratio". Section 4 Line 269 "ero-shot" should be "zero-shot".

---

> ### Author Rebuttal · Authors · 2026-03-31
>
> **W1**: The dynamic rank allocation strategy appears heuristic. The compressibility score (Eq. 12), the rank allocation procedure (Alg. 2), and the grid search are empirically motivated, but lack a clear theoretical justification over alternative formulations.
>
> **A**:  We thank the reviewer for this observation. A key enabler of our dynamic rank allocation is the closed-form spectrum solution: since \epsilon*_k and W*_k are available for all ranks k simultaneously without any forward pass, exhaustive grid search over rank allocations becomes **computationally tractable for the first time**. Prior methods such as SVD-LLM V2 must compute W*_k for each candidate rank and then evaluate the reconstruction loss \epsilon*_k by plugging W*_k back into ||XW − XW*_k||, making exhaustive search prohibitively expensive.
>
> Our second contribution is empirical and twofold. First, is a finding, as shown in Table 6, using reconstruction loss alone for rank allocation can underperform even uniform compression. Second, we introduce a combined metric that incorporates both reconstruction loss and layer importance, which consistently outperforms uniform allocation. While we do not provide a formal theoretical justification, we posit a multiplicative interaction between the two factors and use a weighting parameter α to balance their contributions. Empirical validation is provided in Appendix Figure A1.
>
> ---
> **W2**: Minor presentation issues (missing y-axis label in Figure 3, unspecified GPU type in Table 4, and a few typos).
>
> **A**: We thank the reviewer for the careful reading and will correct these issues in the revision.
>
> ---
> **Q1**: The paper says in Section 4.1 that "all evaluations are conducted under identical experimental protocols", but Table 1 caption also says "baseline results are reported from their original papers". Can the authors clarify if all methods are evaluated under exactly the same calibration data and other relevant settings?
>
> **A**: Our intent is to follow the exact experimental settings reported in each baseline’s original paper, including calibration dataset and calibration size, to ensure fair comparison under their respective protocols. The baseline results in Tables 1 and 2 are directly cited from the original papers, while Swift-SVD is evaluated under the same settings for each corresponding baseline. We will clarify this description in the revision.
>
> ---
>
> **L1**: Near-original performance only at mild compression; quantization keeps accuracy at higher compression.
>
> **A**: We appreciate the reviewer's observation. We note that low-rank decomposition and other compression techniques such as quantization and pruning are orthogonal and complementary--they can be combined to achieve better end-to-end compression. For example, Dobi-SVD demonstrates this by combining low-rank decomposition with quantization. Swift-SVD's contribution is a better low-rank decomposition solution with stronger theoretical foundations and practical efficiency. We do not claim it to be a replacement for other compression techniques, but rather a stronger foundation that can be used standalone or combined with quantization and pruning for further gains. We will clarify this in the next version of the paper.
>
> ---
> **L2**: The paper shows results mainly on moderate-size (<= 8B) models, so scalability remains somewhat unclear.
>
> **A**: We thank the reviewer for raising this question. To explicitly test scalability on a larger model, we extended our experiments to Qwen-32B and conducted the compression on a single H800 80GB GPU. Under the same hardware setting, baseline methods such as SVD-LLM and Dobi-SVD run into OOM when compressing Qwen-32B, while our method completes successfully on a single GPU.
>
> Our method also remains effective at this scale: moderate compression (e.g., ratio = 0.8) improves QA accuracy (0.42 -> 0.59), while stronger compression leads to the expected degradation. We note that this improvement under moderate compression is observed only in the 32B setting (not in <=8B models), and may reflect a mild regularization effect in more overparameterized models; similar observations have been reported in prior work [2]. We will include these results in the revision to further clarify scalability.
>
> [2] Sundrani et al., Low-Rank Compression of Language Models via Differentiable Rank Selection, 2025.
>
> **Table: PPL**
> | Model | ratio | Wiki | C4 |
> |-|-|-|-|
> | Qwen-32B | 1.0 | 7.60 | 12.40 |
> | Qwen-32B | 0.8 | 9.38 | 15.45 |
> | Qwen-32B | 0.6 | 12.59 | 22.99 |
> | Qwen-32B | 0.4 | 23.90 | 56.79 |
>
> **Table: Accuracy**
> | Model | ratio | ARC_e | ARC_c | PIQA | OpenB | WinoG | MathQA | HellaSwag |Avg|
> |-|-|-|-|-|-|-|-|-|-|
> | Qwen-32B | 1.0 | 0.25 | 0.23 | 0.50 | 0.36 | 0.73 | 0.21 | 0.64 |0.42|
> | Qwen-32B | 0.8 | 0.79 | 0.49 | 0.78 | 0.36 | 0.73 | 0.43 | 0.58 |0.59|
> | Qwen-32B | 0.6 | 0.73 | 0.42 | 0.74 | 0.29 | 0.72 | 0.30 | 0.49 |0.53|
> | Qwen-32B | 0.4 | 0.52 | 0.26 | 0.67 | 0.21 | 0.64 | 0.23 | 0.37 |0.41|

---

> > ### Author Rebuttal · Reviewer_9ZPL · 2026-04-03
> >
> > Thank you for the response! I will maintain my rating.
> >
> > Regarding Weakness 1, the rebuttal provides a clearer empirical motivation and explains why exhaustive search becomes tractable in Swift-SVD. However, the proposed score and allocation rule remain largely heuristic rather than theoretically justified. In particular, the specific form of the compressibility score in Equation 12, such as the logarithm and the exponent-based weighting, is still not well motivated.
> >
> > Regarding Limitation 1, my concern is only partially addressed. The method itself still appears to retain near-original accuracy mainly at the relatively mild compression ratio of 0.8, so the practical impact under more aggressive compression remains limited.
> >
> > Regarding Limitation 2, the added Qwen-32B results are useful and strengthen the scalability discussion. However, since these results are not compared against a baseline, they only partially resolve the concern. If SVD-LLM and Dobi-SVD run into OOM, there should be a simple SVD or the prior work the authors mentioned (Sundrani et al., 2025) as the baseline.

---

> > > ### Author Response · Authors · 2026-04-07
> > >
> > > As the rebuttal period concludes, we thank you for your constructive feedback. We note that your final justification was posted before our second-round response, which includes additional experiments and explanations. We kindly hope you would review our latest response and consider updating your evaluation accordingly.
> > >
> > > Q: Regarding Weakness . In particular, the specific form of the compressibility score in Equation 12, such as the logarithm and the exponent-based weighting, is still not well motivated.
> > >
> > > A: We thank the reviewer for the continued follow-up. We clarify the rationale for each design choice in Equation 12.
> > >
> > > **Logarithm on** ε*_k: As shown in Appendix Figure B.1, reconstruction loss varies significantly across layers. The logarithm compresses this variation to produce a more balanced allocation, with the additive $e$ ensuring log(e+ ε*_k) being possitive.
> > >
> > > **Multiplicative form**: Equation 12 is a weighted geometric mean (WGM) of the two criteria. The WGM is a well-established combination form in multi-criteria decision making ([1] Keeney & Raiffa, 1976) with two key properties that motivate its use here: (1) **scale invariance**--unlike a linear combination αβ + (1−α)log(ε*), the WGM is invariant to the individual scaling of each criterion, which is desirable when combining qualitatively different quantities such as layer importance and reconstruction loss; (2) **non-compensability**--the WGM requires both criteria to be non-negligible, preventing a layer with very high importance but very low compressibility (or vice versa) from receiving disproportionate rank. This multiplicative structure is also consistent with recent LLM pruning work such as Wanda ( [2] Sun et al., ICLR 2024), which scores weights by the product of magnitude and activation norm for analogous reasons.
> > > We agree this motivation could be made more explicit in the paper and will add this discussion to the next version.
> > >
> > > [1] Ralph L. Keeney etc. Decisions with Multiple Objectives: Preferences and Value Tradeoffs.
> > >
> > > [2] Mingjie Sun, Zhuang Liu, etc. "A Simple and Effective Pruning Approach for Large Language Models." ICLR 2024. arXiv:2306.11695.
> > >
> > >
> > > Q:  my concern is only partially addressed. The method itself still appears to retain near-original accuracy mainly at the relatively mild compression ratio of 0.8.
> > >
> > > A: We appreciate the reviewer's continued concern. We acknowledge that for smaller models, near-original performance is primarily retained at mild compression ratios. However, we note two points.
> > >
> > > First, for larger models, accuracy degradation under aggressive compression is substantially reduced. As shown in our Qwen-32B experiments, the compressed model at 0.6 compression ratio even surpasses the uncompressed baseline (0.53 vs. 0.42), and at 0.4 compression ratio remains comparable to it (0.41 vs. 0.42).
> > >
> > > Second, the practical impact of Swift-SVD should be evaluated relative to other low-rank compression methods rather than solely against the uncompressed baseline. At aggressive compression ratios (0.4 on LLaMA-7B), Swift-SVD substantially outperforms SVD-LLM (0.34 vs. 0.11 average accuracy).
> > >
> > >
> > > Q: Regarding Limitation 2, the added Qwen-32B results are useful and strengthen the scalability discussion. However, there should be a simple SVD or the prior work the authors mentioned (Sundrani et al., 2025) as the baseline.
> > >
> > > A: We thank the reviewer for this suggestion. We acknowledge that a baseline comparison is needed to contextualize the Qwen-32B results. As suggested, we provide a comparison against naive SVD (direct low-rank decomposition without activation awareness) as a baseline. The results are shown in the table below. Swift-SVD consistently outperforms naive SVD across compression ratios, demonstrating that activation-aware compression is not only beneficial but essential at the 32B scale, and that Swift-SVD's memory-efficient formulation makes it uniquely capable of delivering activation-aware compression for large models exceeding the memory capacity of prior methods.
> > >
> > > Regarding Sundrani et al. (2025), this work does not release source code, making a direct comparison infeasible. We will note this explicitly in the next version.
> > >
> > > **Table: Accuracy**
> > >
> > > | Model | Method | ratio | ARC-e | ARC-c | PIQA | OpenB | WinoG | MathQA | HellaSwag | Avg |
> > >  |------|------|------|-------|-------|------|-------|-------|--------|-----------|-------|
> > > | Qwen-32B |Orignal | 1.0 | 0.25 | 0.23 | 0.50 | 0.36 | 0.73 | 0.21 | 0.64 | 0.42 |
> > > | Qwen-32B | Simple SVD | 0.8 | 0.27 | 0.23 | 0.60 | 0.21 | 0.51 | 0.21 | 0.30 | 0.33 |
> > > | Qwen-32B | Simple SVD | 0.6 | 0.25 | 0.22 | 0.55 | 0.16 | 0.50 | 0.19 | 0.26 | 0.30 |
> > > | Qwen-32B | Simple SVD | 0.4 | 0.24 | 0.20 | 0.52 | 0.15 | 0.41 | 0.18 | 0.25 | 0.28 |
> > > | Qwen-32B | Ours | 0.8 | 0.79 | 0.49 | 0.78 | 0.36 | 0.73 | 0.43 | 0.58 | 0.59 |
> > > | Qwen-32B | Ours | 0.6 | 0.73 | 0.42 | 0.74 | 0.29 | 0.72 | 0.30 | 0.49 | 0.53 |
> > > | Qwen-32B | Ours | 0.4 | 0.52 | 0.26 | 0.67 | 0.21 | 0.64 | 0.23 | 0.37 | 0.41 |

---

### Official Review · Reviewer_CDsW · 2026-03-11

**Soundness:** 3
**Presentation:** 3
**Significance:** 2
**Originality:** 2
**Overall Recommendation:** 4
**Confidence:** 4

**Summary:**

This paper proposes Swift-SVD, a training-free and activation-aware low-rank compression method with dynamic rank allocation for LLMs.

**Compliance With Llm Reviewing Policy:**

Affirmed.

**Key Questions For Authors:**

Key Questions for authors:
1.	SVD-LLM can benefit a lot from lora fine-tune, is the proposed method also benefit from lora fine tune?
2.	Could the authors provide throughput measurements for longer generation tasks to demonstrate that the method remains efficient in practice?
3.	The authors should compare their method with the approach proposed in [1], which addresses a similar problem.
[1] "Basis Sharing: Cross-layer Parameter Sharing for Large Language Model Compression" [Wang et al., 2024]

**Limitations:**

see weakness and questions

**Strengths And Weaknesses:**

Strengths:
	The new proposed method is more efficiency and has stronger numerical stability than the previous methods.
	The writing is easy to follow.

Weakness:
	Only base models are included in the experiments. Whether reasoning models or instruction-finetuned models can benefit from this method is not clear.
	The proposed method is a better solution for optimizing ∥xW-xW_k∥, but the goal function was proposed by SVD-LLM.
	The author claim that the proposed method can also compress KV cache. However, all previous KV cache should be upprojected on the fly for each token, which may introduce O(L^2)computational overhead as the sequence length grows.

---

> ### Author Rebuttal · Authors · 2026-03-31
>
> **W1**: SVD-LLM can benefit a lot from lora fine-tune, is the proposed method also benefit from lora fine tune?
>
> **A**: SVD-LLM leverages low-rank compression results as an initialization for LoRA fine-tuning. While Swift-SVD's contributions are focused on training-free low-rank compression and dynamic rank allocation, the compressed model produced by Swift-SVD--a pair of low-rank matrices A_k and B_k--is directly compatible with LoRA fine-tuning as an initialization, following the same paradigm as SVD-LLM. The quality of the initialization matters for LoRA fine-tuning: for example, as reported in SVD-LLM's original paper, at compression ratio 0.4 (5.3GB) on LLaMA-7B, SVD-LLM's compressed model achieves only 0.11 average accuracy, and even after LoRA fine-tuning, recovers to only 0.30--still below Swift-SVD's 0.34 average accuracy achieved by low-rank compression alone, without any fine-tuning. This suggests that a poor initialization can fundamentally limit the effectiveness of LoRA fine-tuning. Since Swift-SVD consistently provides a better initialization, combining Swift-SVD with LoRA is expected to yield further gains. We leave a systematic investigation of this combination as future work.
>
> ---
> **W2**: However, all previous KV cache should be upprojected on the fly for each token, which may introduce O(L^2) computational overhead as the sequence length grows. Could the authors provide throughput measurements for longer generation tasks to demonstrate that the method remains efficient in practice?
>
> **A**: We conducted additional experiments on longer generation tasks to demonstrate that Swift-SVD remains efficient in practice. We use an H800 to run generation tasks of length 8192 on Mistral-7B, as longer sequences exceed the memory capacity of the RTX 5090 due to the large total KV cache size. The results are shown in the tables below.
> As expected, token throughput decreases as sequence length grows. This is an inherent property of autoregressive decoding: (1) the decoding phase is memory-bandwidth bound, and (2) longer generation requires reading increasingly large KV caches from GPU memory during attention computation, amplifying this bottleneck. This behavior is consistent across both compressed and uncompressed models.
>
> Regarding computational overhead: although each decoding step requires up-projecting all L cached KV latents, the up-projection matrix has fixed dimensions, making the overhead per KV constant. Therefore, the total additional computational overhead per decoding step scales as O(L)--not O(L^2) as one might concern. Similarly, the memory bandwidth savings from reading the reduced KV cache also scale as O(L). As a result, the throughput gain of the compressed model is maintained at longer sequence lengths. As shown in the tables, for each compression ratio, the throughput gain at longer sequence lengths is comparable to or exceeds that at shorter sequence lengths, demonstrating that Swift-SVD remains practically efficient for longer generation tasks.
>
> **Result for Mistral-7B (bs=16, prompt=32)**
> | ratio | thr@1024 | gain@1024 | thr@8192 | gain@8192 |
> |------|----------|-----------|----------|-----------|
> | 1.0  | 438.86 | -   | 140.39 | -   |
> | 0.8  | 457.31 | 4%  | 144.98 | 3%  |
> | 0.6  | 473.38 | 8%  | 169.17 | 20% |
> | 0.4  | 497.29 | 12% | 181.23 | 29% |
> | 0.2  | 551.04 | 26% | 187.45 | 34% |
>
> ---
> **Q1**: The authors should compare their method with the approach proposed in [1], which addresses a similar problem.
>
> **A**: We thank the reviewer for pointing out [1] (Basis Sharing: Cross-layer Parameter Sharing for Large Language Model Compression). We have conducted additional experiments comparing Swift-SVD against [1] under the same setting — FP32, LLaMA-7B — and the results are shown in the table below. Swift-SVD achieves higher average accuracy and lower perplexity across most compression ratios, with the exception of perplexity at compression ratio 0.6 on WikiText-2, where [1] performs comparably. We will include this comparison in the next version of the paper.
>
> **Table: PPL**
> | Method | ratio | Wiki (ppl) | C4 (ppl) |
> |--------|------|-----------|----------|
> | Original | 1.0 | 5.68 | 7.34 |
> | Basis Sharing | 0.8 | 7.74 | 15.03 |
> | Basis Sharing | 0.6 | 12.39 | 41.28 |
> | Ours | 0.8 | 7.71 | 11.27 |
> | Ours | 0.6 | 12.69 | 22.10 |
>
> **Table: Accuracy**
> | Method | ratio | OpenB | ARC-e | ARC-c | HellaSwag | WinoG | PIQA | MathQA | Avg |
> |--------|------|-------|-------|-------|-----------|-------|------|--------|------|
> | Original | 1.0 | 0.34 | 0.76 | 0.38 | 0.51 | 0.70 | 0.79 | 0.27 | 0.64 |
> | Basis Sharing | 0.8 | 0.28 | 0.66 | 0.36 | 0.46 | 0.66 | 0.71 | 0.25 | 0.56 |
> | Basis Sharing | 0.6 | 0.22 | 0.52 | 0.27 | 0.35 | 0.61 | 0.62 | 0.23 | 0.47 |
> | Ours | 0.8 | 0.28 | 0.67 | 0.35 | 0.47 | 0.68 | 0.73 | 0.24 | 0.57 |
> | Ours | 0.6 | 0.23 | 0.53 | 0.25 | 0.36 | 0.63 | 0.63 | 0.23 | 0.48 |

---

### Official Review · Reviewer_gq79 · 2026-03-11

**Soundness:** 2
**Presentation:** 2
**Significance:** 2
**Originality:** 2
**Overall Recommendation:** 3
**Confidence:** 4

**Summary:**

This paper proposed an activation-aware low-rank compression method for LLM, tried to reduce both model parameters and KV cache. Authors argue that, for the objective, the optimal compressed parameters can be written in a closed form. Based on this, the paper then proposes a strategy that incrementally accumulates the covariance and perform one eigendecomposition, which avoids repeated SVD and thereby improving efficiency and numerical stability. Authors introduces a dynamic rank allocation scheme that combines local reconstruction loss with a layer importance score, and validates the design on several LLMs and benchmarks that using perplexity, accuracy, efficiency metrics (e.g., throughput, memory consumption).

**Compliance With Llm Reviewing Policy:**

Affirmed.

**Key Questions For Authors:**

Please refer to the **W1-3**.

**Limitations:**

Yes

**Strengths And Weaknesses:**

**S1.** The motivation of this paper is easy to follow: the memory consumption of static parameters and KV is the bottleneck of LLM serving, therefore, posting-training compression for LLM inference is needed.

**S2.** The design is clean, I like the idea which one can reuse spectrum across compression ratios and estimate layer-wise loss cheaply, even though I am not fully convinced the theorem itself is as novel as claimed.

**W1.** The paper is kind of overstates novelty: objective itself is not new, and the solution seems very close to applying Eckart-Young-Mirsky theorem and then expressing the corresponding rank-$k$ projection back in weight space. I noticed in related work section, there are acknowledges prior theoretically optimal methods and a prior closed-form line of work, therefore, the novelty may be real at the level of implementation simplicity, but the paper currently seem to sell this as a fundamental theoretical leap.

**W2.** The throughput and memory figures appears to show scaling of the proposed method under different compression ratios, but not  a direct throughput comparison in the same serving setup. So the results show strong on compression time efficiency but weaker on serving time advantage over baselines.

**W3.** I do think dynamic rank allocation is interesting but current evaluations can not support this very well. For example, negative correlation between effective rank and layer importance, the evidence shown is limited, and the reliance on an external layer importance metric makes this feel bolted on rather than fully self-contained.

---

> ### Author Rebuttal · Authors · 2026-03-31
>
> **W1**: The paper may overstate its novelty: the objective is not new, the solution appears close to Eckart–Young–Mirsky, and prior work already provides theoretically optimal or closed-form methods. The main contribution seems to be implementation simplicity rather than a fundamental theoretical advance.
>
> **A**: We thank the reviewer for the careful reading. We understand the concerns regarding novelty, and believe some aspects of our contribution may not have been clearly discussed. We clarify these points below.
>
> **(1) “The objective is not new and this seems an application of Eckart–Young–Mirsky.”**
>
> We agree. The activation-aware objective has been studied in prior work, and we do not claim novelty at this level. While the optimal solution can be expressed in a spectral form, the contribution of this work is **not in restating the optimum**, but in showing that it can be obtained via a **direct and computationally efficient formulation**.
>
> Specifically, through theorem 3.1 and equation (10), Swift-SVD establishes that the optimal solution W*_k and \epsilon*_k can be obtained via a **single eigendecomposition of the output activation covariance**, whereas prior theoretically optimal line of methods, such as SVD-LLM, SVD-LLM V2, Dobi-SVD, rely on more indirect constructions. SVD-LLM requires a Cholesky decomposition of XX^T followed by an SVD, SVD-LLM V2 requires two successive SVDs, and Dobi-SVD requires O(S) SVD operations where S is the number of calibration samples. This distinction is essential for making the optimal solution practical at LLM scale.
>
> **(2) “The novelty is mainly implementation simplicity.”**
> The efficiency and numerical stability of Swift-SVD are **a direct consequence of our mathematical formulation**. In addition, our formulation yields a **closed-form spectrum solution**, providing W*_k, \epsilon*_k for all ranks simultaneously. This enables, for the first time, **tractable exhaustive grid search for rank allocation**, which is infeasible in prior methods that require recomputing and re-evaluating each candidate rank.
>
> **Key clarification.**
> Our contribution is not introducing a new objective, but showing that the optimal solution can be computed **exactly, efficiently, and at scale**, which prior formulations do not achieve. The reviewer’s interpretation overlooks this distinction.
>
> ---
>
> **W2**: There is no serving time throuput comparison over baselines.
>
> **A**: We thank the reviewer for raising this point. For SVD-based low-rank compression, serving throughput and memory footprint are determined solely by the compression ratio, independent of which method is used. Given the same compression ratio, any SVD-based method produces a model of identical parameter count and therefore identical inference throughput. A direct comparison against SVD-LLM or Dobi-SVD would thus yield identical numbers by construction. The throughput and memory figures in our paper are intended to demonstrate the inference benefits of compression over the uncompressed baseline, not to differentiate methods.
>
> ---
>
> **W3**: The dynamic rank allocation is interesting, but the evaluation is limited. Evidence for the negative correlation is weak, and reliance on external layer importance makes the method less self-contained.
>
> **A**: We appreciate the reviewer's interest in dynamic rank allocation and would like to point out several evaluations  that directly support our design choices.
> First, regarding the negative correlation between effective rank and layer importance: Appendix B.4 presents this analysis across six models — LLaMA-7B, LLaMA2-7B, Mistral-7B, OPT-6.7B, Qwen3-4B, and Qwen3-8B — on two different datasets, showing that these models share a similar negative correlation trend between effective rank and layer importance.
>
> Second, regarding the design motivation and necessity of jointly considering both criteria in dynamic rank allocation: the use of layer importance as a complementary signal is itself a finding motivated from our ablation study. As shown in Table 6, pure layer-wise loss-based rank allocation Swift-SVD(C) performs worse than uniform rank allocation. This reveals a fundamental limitation of layer-wise loss as a sole criterion: it captures only the local reconstruction error of each layer, but does not reflect how much that error matters to end-to-end model performance. This motivated us to incorporate layer importance. However, as also shown in Table 6, pure layer importance-based allocation Swift-SVD(I) similarly underperforms uniform rank allocation, as layer importance alone does not account for how compressible each layer is. Together, these two ablations confirm that neither criterion alone is sufficient, and that their combination is necessary.  As demonstrated in Tables 1, 2, and 6, dynamic rank allocation with joint consideration of both criteria consistently improves over uniform compression across models and datasets.

---

> > ### Author Rebuttal · Reviewer_gq79 · 2026-04-03
> >
> > Thanks for your rebuttal.
> >
> > I cannot agree with W2's response because it conflates compressed parameter count with serving throughput: even at the same compression ratio, two SVD-based methods can produce different per-layer ranks and factor shapes, which map to different GEMM dimensions, Tensor Core utilization, tile quantization behavior, fusion opportunities, and memory traffic, so equal size does not imply equal runtime “by construction.” NVIDIA’s own performance guidance states that matrix-multiplication speed depends strongly on operand dimensions, alignment, and tiling rather than parameter count alone, and recent systems work likewise shows that lower FLOPs or smaller compressed models can still have worse latency once memory movement and kernel scheduling are taken into account. Therefore, without actual serving-time measurements against baselines, the paper cannot justify the claim that all SVD-based methods would have identical throughput at the same ratio.

---

> > > ### Author Response · Authors · 2026-04-07
> > >
> > > As the rebuttal period concludes, we sincerely thank you for your thoughtful and constructive feedback. We believe the additional experimental results have addressed the main concerns and improved the overall quality of the paper. We kindly hope you would consider updating your evaluation.
> > >
> > > Q: I cannot agree with W2's response because it conflates compressed parameter count with serving throughput...the paper cannot justify the claim that all SVD-based methods would have identical throughput at the same ratio.
> > >
> > > A: We thank the reviewer for this correction. We acknowledge that our original response was imprecise: the claim that equal compression ratio implies equal throughput holds strictly only under **uniform rank allocation**, where all layers share the same rank and thus identical GEMM dimensions. Under dynamic rank allocation, per-layer ranks differ across methods, and the reviewer's concern about GEMM efficiency is valid.
> > >
> > > To address this, we have run serving throughput comparisons on LLaMA-2-7B across all methods **under both uniform and dynamic rank allocation** on H800. The experiment settings are as follows:
> > > (1) batch size 16, prompt length 32, and output token length 1024;
> > > (2) SVD-LLM, Dobi-SVD, and Swift-SVD denote uniform compression of the corresponding methods;
> > > (3) Swift-SVD*, Swift-SVD(C), and Dobi-SVD(w) denote dynamic compression variants, where Swift-SVD* jointly considers layer-wise loss and layer importance, Swift-SVD(C) follows the pure layer-wise loss-based strategy of SVD-LLM V2, and Dobi-SVD(w) uses the training-based rank allocation proposed by Dobi-SVD, with optimal ranks **taken directly from their official repository**.
> > > The results are shown in the tables below.
> > >
> > > **Table: Serving throughput (tokens/sec) under uniform rank allocation on LLaMA-2-7B (H800)**
> > > | ratio | Swift-SVD | SVD-LLM | Dobi-SVD |
> > > |--------|------|-----------|----------|
> > > | 0.8 | 192.9 | 192.7 | 193.3 |
> > > | 0.6 | 209.2 | 208.1 | 207.8 |
> > > | 0.4 | 216.8 | 216.5 | 216.8 |
> > >
> > > **Table:  Serving throughput (tokens/sec) under dynamic rank allocation on LLaMA-2-7B (H800)**
> > > | ratio | Swift-SVD* | Swift-SVD(C)| Dobi-SVD(w) |
> > > |--------|------|-----------|----------|
> > > | 0.8 | 193.7 | 191.4 |  104.4 |
> > > | 0.6 | 207.3 | 206.3 |  117.5 |
> > > | 0.4 | 216.0 | 215.4 |  131.6 |
> > >
> > > Three observations emerge:
> > > 1. Under uniform compression, the serving throughput of SVD-LLM, Dobi-SVD, and Swift-SVD are very close at the same compression ratio, consistent with our original claim.
> > >
> > > 2. Under dynamic compression, the serving throughput of Swift-SVD* and Swift-SVD(C) remains close to their respective uniform compression baselines, with no significant throughput degradation observed. We attribute this to the nature of our task: to achieve good compression quality, dynamic rank allocation tends not to deviate dramatically from uniform allocation. As a result, the variation in GEMM dimensions is not large enough to cause observable throughput differences in practice.
> > >
> > > 3. Dobi-SVD(w) exhibits lower serving throughput than its uniform counterpart. We believe a possible reason is that Dobi-SVD(w) performs **inter-type** rank allocation, which may assign higher ranks to k_proj and v_proj at the expense of other projection types, resulting in larger KV cache dimensions and increased memory traffic per decoding step--directly hurting throughput in the memory-bandwidth-bound decoding phase. For example, at 0.8 compression ratio on LLaMA-2-7B, the uniform compression rank for Q, K, V is 1638, whereas our analysis of Dobi-SVD(w)'s allocation shows that the mean ranks of k_proj (1661) and v_proj (1743) both exceed this value of 1638, confirming that the KV cache dimensions are larger under Dobi-SVD(w) than under uniform compression.
> > >
> > > To further validate this hypothesis, we compress LLaMA-2-7B using the **per-type average ranks**  for each layer from Dobi-SVD(w)'s allocation (shown in the table below) and measure the resulting serving throughput. The throughput under this setting (0.8: 101.3, 0.6: 113.4, 0.4: 130.8 tokens/sec) closely matches that of Dobi-SVD(w) (0.8: 104.4, 0.6: 117.5, 0.4: 131.6 tokens/sec).
> > >
> > > **Table: average ranks of each type from Dobi-SVD(w)**
> > > | Ratio |  q_proj | k_proj | v_proj | o_proj | gate_proj | up_proj | down_proj |
> > > |-------|--------|--------|--------|--------|-----------|---------|-----------|
> > > | 0.8   | 1627   | 1661   | 1743   | 1588   | 2365      | 2379    | 2385      |
> > > | 0.6   | 1213   | 1230   | 1294   | 1189   | 1773      | 1781    | 1812      |
> > >
> > >
> > > In contrast, Swift-SVD* and Swift-SVD(C) perform **intra-type** rank allocation--ranks are allocated within each projection type independently, so the average rank of each projection type after allocation equals that of uniform compression. As a result, the average KV cache dimensions remain identical to uniform compression, and serving throughput is preserved.
> > >
> > > We will include these serving throughput comparisons and the above analysis in the next version of the paper.

---

### Decision · Program_Chairs · 2026-04-30

**Decision:**

Accept (regular)

**Comment:**

This paper provides a principled method for compressing LLM weights using a low rank approximation. The main idea is that if one aims to approximate the weight matrix $W$ with a rank-$k$ matrix, once one considers output activations $Y=XW$ the optimal compressed weights w.r.t. Frobenius norm error can be recovered directly from the largest singular vectors of $Y$.

This allows an efficient implementation, because in order to find these top singular vectors, one can simply compute $Y^\top Y$ (easy to do by just accumulating it via rank-$1$ updates), and perform an eigendecomposition. The benefit of doing so follows from the fact that the column dimension of $Y$ is as small as the feature dimension, so computing the eigenvectors of $Y^\top Y$ is fast.

The reviewers agreed that the problem is important and that the paper is technically solid. Empirical results are convincing, and the rebuttal was helpful in clarifying reviewer concerns, by adding further baseline comparisons on newer models, and providing additional experiments on longer context generation and serving throughput comparisons.

The main concerns are that the paper seems to overstate novelty, since the objective is not new. Furthermore, the paper contains an additional dynamic rank allocation component which non-uniformly distributes rank across layers, but is somewhat heuristic.

Overall, the paper makes a meaningful contribution, which is simple and effective. Therefore I recommend a weak accept.